# A Nodal enhanced micropeptide NEMEP regulates glucose uptake during mesendoderm differentiation of embryonic stem cells

Haipeng Fu[1], Tingyu Wang[1], Xiaohui Kong[1], Kun Yan[2], Yang Yang[1,3,4], Jingyi Cao[2], Yafei Yuan[5], Nan Wang[6], Kehkooi Kee[6], Zhi John Lu[1,3] & Qiaoran Xi[1,4✉]

TGF-β family proteins including Nodal are known as central regulators of early development in metazoans, yet our understanding of the scope of Nodal signaling's downstream targets and associated physiological mechanisms in specifying developmentally appropriate cell fates is far from complete. Here, we identified a highly conserved, transmembrane micropeptide— NEMEP—as a direct target of Nodal signaling in mesendoderm differentiation of mouse embryonic stem cells (mESCs), and this micropeptide is essential for mesendoderm differentiation. We showed that NEMEP interacts with the glucose transporters GLUT1/GLUT3 and promotes glucose uptake likely through these interactions. Thus, beyond expanding the scope of known Nodal signaling targets in early development and showing that this target micropeptide augments the glucose uptake during mesendoderm differentiation, our study provides a clear example for the direct functional impact of altered glucose metabolism on cell fate determination.

[1] MOE Key Laboratory of Protein Sciences, School of Life Sciences, Tsinghua University, Beijing 100084, China. [2] Tsinghua-Peking Center for Life Sciences, School of Life Sciences, Tsinghua University, Beijing 100084, China. [3] MOE Key Laboratory of Bioinformatics, Center for Synthetic and Systems Biology, School of Life Sciences, Tsinghua University, Beijing 100084, China. [4] Joint Graduate Program of Peking-Tsinghua-NIBS, Tsinghua University, Beijing 100084, China. [5] State Key Laboratory of Membrane Biology, Beijing Frontier Research Center for Biological Structure, Beijing Advanced Innovation Center for Structural Biology, Tsinghua-Peking Joint Center for Life Sciences, School of Life Sciences, Tsinghua University, Beijing 100084, China. [6] Center for Stem Cell Biology and Regenerative Medicine, Department of Basic Medical Sciences, School of Medicine, Tsinghua University, Beijing 100084, China. ✉email: xiqiaoran@mail.tsinghua.edu.cn

Glucose is a prominent carbohydrate in the metabolism of most organisms, and an increasing number of developmental biology studies are lending support to the hypothesis that glucose metabolic processes per se functionally contribute to cell fate determination in early embryogenesis[1–3]. Mouse embryos start to rely on glucose as a major energy source at the stage of compaction and blastocyst formation (around E4–E5)[4], and there is genetic evidence confirming that active glycolysis is required for gastrulation[5]. Clear phenotypes support that two major glucose transporter proteins (GLUT1 and GLUT3) are essential for early development: $Glut1^{-/-}$ mice are embryonic lethal at E10[6], and deletion of GLUT3 results embryonic lethality at the gastrulation stage[7]. And studies have shown that increasing the expression of GLUT1 and GLUT3 in the pluripotent inner cell mass (ICM) increases glucose uptake, which in turn triggers enhancement of glycolytic flux for ATP generation and a subsequent increase in lactate excretion[8–10]. Despite these exciting observations, to date very little is known about nature of any regulatory system(s) that direct the activation of glucose uptake to meet the specific needs of distinct cell populations during early development.

Transforming growth factor β (TGF-β) signaling represents a centrally important regulatory influence on early development, and members of the TGF-β family are also known to function in widespread and diverse roles in tissue homeostasis, wound healing, immunity, and metabolism[11–15]. These proteins mediate their multifunctional effects in cells by eliciting transcriptional responses on many target genes, specifically through the receptor-activated SMADs (R-SMADs). R-SMADs—SMAD2 and SMAD3 for TGFβ, Nodal, Activin, and myostatin signaling—translate TGF-β pathway signals into the specific transcriptional programs in specific cell contexts, doing so via their interactions with SMAD4 and other transcriptional activators, coactivators/corepressors, and/or epigenetic regulators at enhancer elements in specific genes[16–21]. Nodal signaling initiates gastrulation in vivo and promotes embryonic stem cells (ESCs) to differentiate towards mesendoderm fates in vitro; these regulatory processes are mediated by the transcriptional upregulation of so-called "mesendoderm lineage determining transcription factors" (LDTFs)[22–24]. Further, the interactions of SMAD2 and SMAD3 with FOXH1 and TRIM33 are known to regulate essential transcriptional programs to direct mesendoderm differentiation[25–27].

The insights about transcriptional regulation of mesendoderm LDTFs have advanced our understanding about how Nodal signaling regulates mesendoderm differentiation, but it is far from clear that the only impacts of Nodal signalling on early development are mesendoderm LDTFs. Long non-coding RNAs (lncRNAs) are defined as polyadenylated RNA molecules which are longer than 200 nucleotides, are weakly conserved, and are transcribed by RNA polymerase II[28–30]. Studies have shown that lncRNAs exert diverse functions in biological processes, including development, metabolism, immunity, and disease[31–36]. Our understanding of lncRNAs is expanding rapidly, and it is now known that some cytosol-localized lncRNAs can be translated into functional micropeptides in vivo[37].

Here, we identified a Nodal signaling direct target gene, Gm11549, which was originally annotated as a lncRNA. To our considerable surprise, we later found that Gm11549 actually encodes a highly conserved 63 amino acid single-pass transmembrane micropeptide, NEMEP (Nodal Enhanced MEsendoderm Peptide). Through multiple follow-up functional investigations, we discovered that NEMEP interacts with two glucose transporters (GLUT1 and GLUT3) and promotes glucose uptake. Interestingly, NEMEP is specifically accumulated in the primitive streak of mouse embryo at E7.0, and depletion of NEMEP causes two strong phenotypes: dramatic impairment of mesendoderm differentiation and a significant decrease in glucose uptake in early differentiated mESCs. Hence, our study establishes an additional role of Nodal signaling in mesendoderm differentiation beyond its induction of LDTFs and shows how a Nodal-regulated micropeptide functions to augment glucose uptake into specific subpopulations of mesendodermal cells.

## Results

**Gm11549 is a direct target gene of Nodal signaling.** Mouse embryonic stem cells (mESCs) are widely used to experimentally recapitulate early embryonic development, for example with in vitro assays examining embryoid body (EB) formation. The LIF/STAT3 (Leukemia Inhibitory Factor/Signal Transducer and Activator of Transcription-3) signaling pathway promotes self-renewal and blocks the differentiation of mouse ESCs. LIF removal from culture media is the first step in a common protocol for inducing the differentiation of mESCs[38–40]. This system is developmentally informative because mESCs have the potential to differentiate into all three germ layers upon selective exposure to appropriate culture conditions[41]. mESCs express ALK4, ALK7, ActR-II, and ActR-IIB and produce autocrine Nodal[42]. We set out to explore lncRNAs responsive to Nodal signaling during mesendoderm differentiation. Activin A was used as a substitute for Nodal in our study because it is easier-to-obtain and because these two protein ligands act through the same receptors; one notable distinction is that Nodal requires the co-receptors Cryptic and Cripto; Activin A does not[43].

A previous RNA-seq study of day 2.5 EBs treated with either the Nodal signaling agonist Activin A (hereafter, Activin) or ALK4/5/7 inhibitor SB431542 globally characterized the mesendoderm LDTFs genes induced by Nodal/Activin signaling[44,45]: these include the homeobox protein gene *Mixl1*, the T-box transcription factor *Brachyury* gene (*Bra*; also known as *T*), the homeobox protein gene *Goosecoid* (*Gsc*), and the hepatocyte nuclear factor 3-β gene *Foxa2*. Note that each of these LDTFs is gradually induced in EBs between day 2 and day 4. Here, we re-analyzed the Wang et al. (2017) dataset[44] with a particular focus on identifying Nodal/Activin signaling target lncRNAs, and thus identified 62 lncRNAs whose expression in SB431542-treated cells was increased or decreased by at least 2-fold upon Activin addition (Fig. 1a and Supplementary Fig. 1a).

Among the Nodal/Activin responsive lncRNAs, we subsequently filtered for candidates using two criteria: strong expression in EBs at day 3 and day 4 and the presence of a predicted SMAD2 or SMAD3 binding site in their promoter regions[25,44]. This identified one candidate, *Gm11549* (Fig. 1b, Supplementary Fig. 1b, c), and we characterized the *Gm11549* transcript using 3′end RACE from EBs at day 3, and the *Gm11549* transcript we found in EBs differs from Genbank and Refseq (Fig. 1b). ChIP-seq analysis showed that the *Gm11549* locus does have TRIM33, SMAD2 or SMAD3, SMAD4, and FOXH1 occupancy at its promoter region in Activin-treated EBs at day 2.5 and does not have SMAD2 or SMAD3 binding in SB431542-treated cells, suggesting that *Gm11549* expression is responsive to Nodal/Activin signaling (Fig. 1b). Pretreatment of cycloheximide, the protein synthesis inhibitor, to the EBs did not block Activin response of *Gm11549* demonstrating that *Gm11549* is a primary transcriptional target of Nodal/Activin signaling, but is not regulated by the Nodal/Activin signaling downstream target genes (Fig. 1c).

We then examined the distribution of *Gm11549* transcripts throughout an in vitro differentiation time course of EBs and found that *Gm11549* is induced at a similar time as the known mesendodermal marker genes *Mixl1*, *Gsc*, *T*, and *Foxa2* (Fig. 1d). Consistently, we analyzed published in vivo data[46,47] for mouse

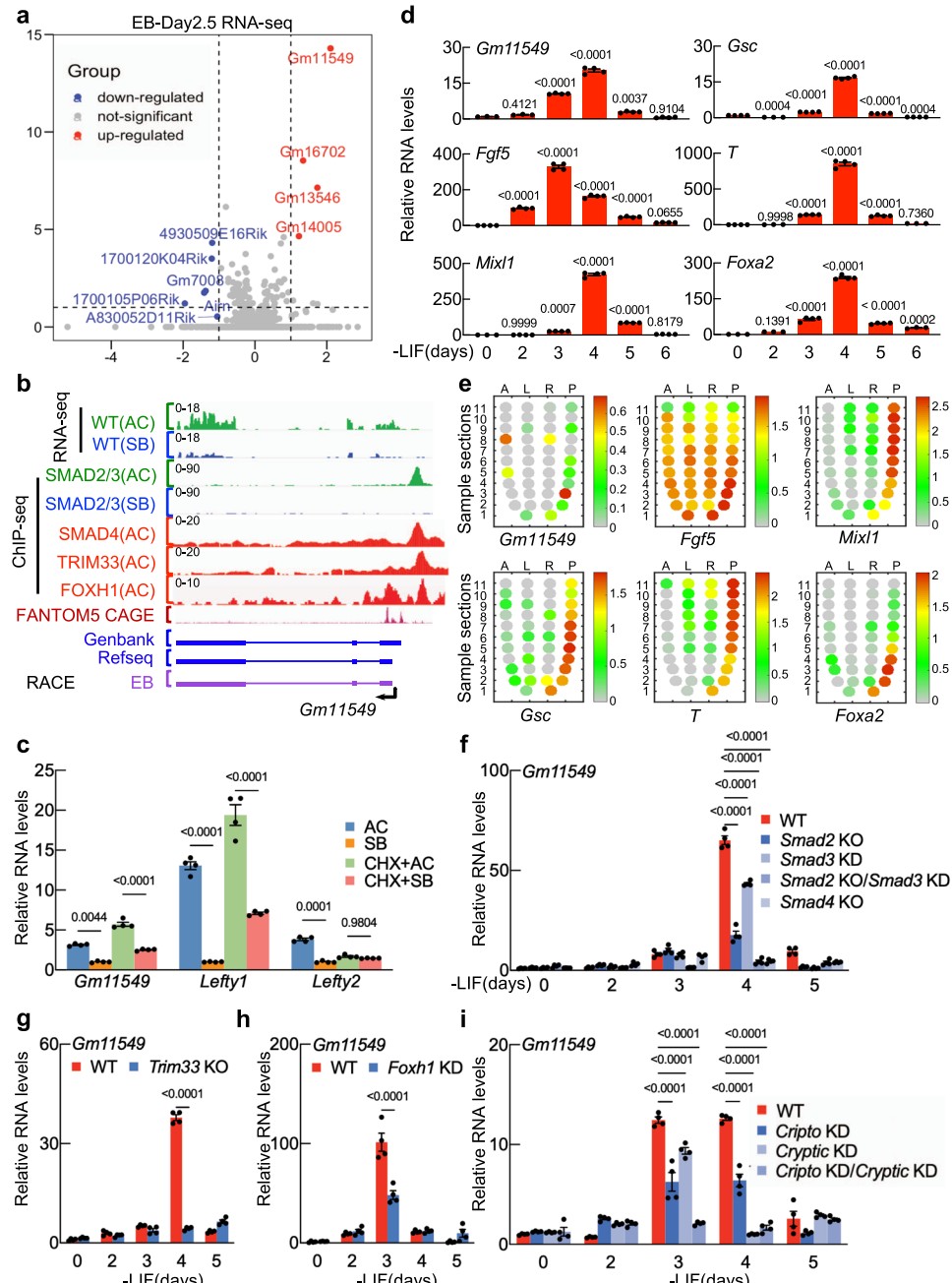

**Fig. 1 Gm11549 is a direct target gene of Nodal signaling in mesendoderm differentiation of mESCs. a** Volcano plot showing differentially accumulated long noncoding RNAs from transcriptome datasets of day 2.5 murine EBs treated with either Activin A or SB431542 for 2 h ($n = 2$ biologically independent samples). Blue, long noncoding transcripts downregulated in Activin A -versus SB431542-treated EBs (fold change < 0.5, $p < 0.05$); Red, long noncoding transcripts upregulated in Activin A- versus SB431542- treated EBs (fold change > 2, $p < 0.05$) (GEO: GSE115169). **b** Gene track view of RNA-seq for Activin A (AC)- and SB431532 (SB)—treated day 2.5 EBs for 2 h; SMAD2/3 ChIP-seq, SMAD4 ChIP-seq, TRIM33 ChIP-seq, and FOXH1 ChIP-seq in AC- and SB- treated day 2.5 EBs for 2 h (GSE70486, GSE125116); FANTOM5 CAGE at the *Gm11549* locus. Three forms of *Gm11549* transcript (Genbank, Refseq, and our RACE from days 3 EBs) are shown. **c** qPCR analysis of *Gm11549, Lefty1,* and *Lefty2* expression in Activin A (AC)—or SB431542 (SB) - treated day 3 EBs for 2 h with or without pretreatment with cycloheximide (CHX) for 1 h. **d** qPCR analysis of the indicated transcripts in day 0 to day 6 EBs of in vitro differentiation. *Fgf5* is a primary ectoderm marker. *Mixl1, Gsc,* and *T* are markers for the primitive streak. *Foxa2* is a marker for the definitive endoderm. **e** Adapted from[47]: Corn plots showing the spatial pattern of *Gm11549* expression in E7.0 mouse embryos. A: anterior; P: posterior; L: left lateral; R: right lateral. **f** qPCR analysis of *Gm11549* expression in day 0 to day 5 EBs from WT or *Smad2* KO, *Smad3* KD, *Smad2* KO/*Smad3* KD, and *Smad4* KO cells. **g** qPCR analysis of *Gm11549* expression in day 0 to day 6 EBs from WT or *Trim33* KO cells. **h** qPCR analysis of *Gm11549* expression in day 0 to day EBs from WT or *Foxh1* KD cells. **i** qPCR analysis of *Gm11549* expression in day 0 to day 5 EBs from in WT, *Cripto* KD, *Cryptic* KD, *Cripto* KD/*Cryptic* KD cells. **c**, **d**, **f–i**: Data are the mean ± S.E.M, $n = 4$ biologically independent samples. $P$ values were determined by one-way (**d**) with Dunnett's corrections or two-way ANOVA with Tukey's corrections (**c**, **f**, **i**) or with Sidak corrections (**g**, **h**), and data are representative of three independent experiments with similar results. Source data are provided as a Source Data file.

embryos and found that, *Gm11549* transcripts accumulate in the posterior primitive streak of mid-gastrulation embryos (~E7.0) (Fig. 1e and Supplementary Fig.1d). Notably, in adult mouse tissues, *Gm11549* is highly abundant and enriched in the brain compared to other organs (Supplementary Fig. 1e).

To determine which factors in Nodal/Activin signaling induce *Gm11549* transcription, we examined *Gm11549* levels and responsivity to Activin in series of gene knockout mESCs. Briefly, both *Gm11549* transcription and responsivity of *Gm11549* to Nodal/Activin are dependent on SMAD2, SMAD3, and SMAD4 individually, as both phenotypes were impaired in *Smad2* or *Smad3* or *Smad4* depletion cells or *Smad2* KO and *Smad3* KD cells (Fig. 1f, and Supplementary Fig. 1f, g). We also found that TRIM33— a Nodal signaling-specific chromatin reader that associates with H3K9me3 and H3K18ac dual marks to regulate the expression of mesendoderm LDTFs[21,25]—is essential for *Gm11549* induction during mesendoderm differentiation and *Gm11549* transcription was not induced by Nodal/Activin stimulation in *Trim33* null cells (Fig. 1g, and Supplementary Fig. 1h, i). Finally, we found that *Gm11549* transcriptional induction during mesendoderm differentiation requires the known early development regulator FOXH1 (Fig. 1h, and Supplementary Fig. 1j, k). In addition, depletion of co-receptors of Nodal, Cryptic, and Cripto (Supplementary Fig. 1l), dramatically impairs *Gm11549* expression during mesendoderm differentiation, as well as *Mixl1* (Fig. 1i, Supplementary Fig. 1m), suggesting that *Gm11549* is indeed a Nodal signaling target gene. Collectively, these results establish that Nodal signaling components directly induce the transcription of *Gm11549* during mesendoderm differentiation and demonstrate that *Gm11549* is specifically expressed in the primitive streak.

**Gm11549 encodes a transmembrane micropeptide: NEMEP.** Portions of the *Gm11549* genomic sequence are highly conserved between human and mouse (Fig. 2a); specifically, exon1 (224–405) and exon3 (1121–1446) share 87% and 75% identity respectively at nucleic acid sequence level, whereas there is no obvious conservation for introns. Moreover, the neighboring genes on both sides of the *Gm11549* locus are conserved in both species (Fig. 2a). RNA-FISH (Fluorescence in situ hybridization) and fractionation-based analyses revealed that *Gm11549* transcript is mainly localized in the cytoplasm (Supplementary Fig. 2a, b). Recent studies reported that some cytosol-localized lncRNAs can be translated into functional micropeptides in vivo[37], which prompted us to further investigate whether *Gm11549* may have the potential to be translated into a protein product.

First, we identified 4 putative ORFs in a sense (+) orientation, with initiating ATG codon (Fig. 2b). After cloning these ORFs, we examined the translational potential of these ORFs by transiently expressing them as C-terminal tagged fusions bearing FLAG epitopes in HEK293T cells. Only the ORF1-FLAG was detected by western blotting (Fig. 2c); mutagenesis at the ATG codon abolished ORF1-FLAG expression (Supplementary Fig. 3a, b); note that ORF1 encodes a 63 amino acid micropeptide that is conserved among mammals (Fig. 2d). Moreover, polysome profiling analysis of EBs at day 3 showed that *Gm11549* is strongly associated with the polysome-like coding gene *Gsc*, whereas non-coding RNA *H19* is associated with the monosomes (Fig. 2e). These data provided additional evidence that *Gm11549* has translation potential.

Further, peptides from ORF1 were detected in HEK293T cells transiently expressing the ORF1-FLAG product by pull-down with the FLAG antibody followed by mass-spectrometry analysis (Fig. 2f). Moreover, we raised a polyclonal rabbit antibody against

a 25-residue region of the ORF1 gene product (Fig. 2d), an analysis of adult mouse brain samples with pull-down using the polyclonal antibody again detected the *Gm11549* ORF1 protein (Fig. 2g). We generated ORF1-FLAG knock-in mESCs using CRISPR/Cas9 (Supplementary Fig. 3c), and a FLAG-tagged protein with a molecular weight of about 7KD was detected in these cells by western blotting (Fig. 2h). These results together establish that the micropeptide from *Gm11549* ORF1 is indeed translated in vivo. We refer to this Nodal signaling target gene product as Nodal Enhanced MEsendoderm microPeptide (NEMEP).

The human ortholog of *Gm11549 is TMEM155*, which was annotated as a coding gene for a 130 aa gene product; note that there are to date no reports of experimental confirmation of the existence of these 130 aa proteins. In experiments to examine the translation product of TMEM155, we performed similar assays as with *Gm11549* and found that *TMEM155* likewise encodes a 63 aa peptide ("hNEMEP", sharing 93.6% identity with mNEMEP) (Supplementary Fig. 3d).

The protein topology prediction analyses indicated that NEMEP is a single-pass transmembrane protein (TMHMM Server 2.0) (Supplementary Fig. 3e), and NEMEP has a predicted alpha-helix domain between residues 7 and 29 (Supplementary Fig. 3f). Immunostaining of ectopically expressed, FLAG-tagged NEMEP in mouse ES cells revealed that NEMEP was localized at the plasma membrane (Fig. 2i), and further, fractionation-based analysis and immunoblotting of mESCs expressing NEMEP-GFP indicated strong accumulation of an NEMEP-GFP fusion protein in the membrane fraction (Fig. 2j).

**NEMEP is required for mesendoderm differentiation.** We next used CRISPR/Cas9 to generate a Gm11549-NEMEP frameshift mutants (hereafter, *Nemep* KO): we picked two frameshift mutant colonies, one with an 8nt deletion (KO-1) and another with a 77nt deletion (KO-2); neither of these mutant *Gm11549* loci produced the NEMEP peptide in cells (Supplementary Fig. 4a–c). However, note that the in vitro differentiation induced elevation of *Gm11549* RNA transcription was not altered by either of these mutations (Fig. 3a). The absence of NEMEP in EBs dramatically impaired the expression of well-known mesendoderm LDTFs including *Gsc, Mixl1, T, Eomes, Foxa2,* and *Sox17* (Fig. 3a). Consequently, the expression of the late mesoderm marker *Nkx2–5* was significantly decreased in the *Nemep* KO EBs (Fig. 3a). Strikingly, although *Gm11549* transcription is induced during ectoderm differentiation, we found that this RNA molecule is not required for ectoderm differentiation (Fig. 3b and Supplementary Fig. 4d). We also found that the phosphorylation level of SMAD2 and SMAD3 at C-terminal tail is not affected in *Nemep* KO cells compared to wild type cells (Fig. 3c), indicating that NEMEP deletion does not interfere with the activity of known Nodal/Activin signaling receptors. Moreover, we found that neither depletion of exon2 or exon3 has impact on mesendoderm differentiation, excluding any regulatory impacts from *Gm11549* region other than the 5′ UTR and coding region for 63 aa NEMEP (Supplementary Fig. 4a and 5a, b). This data also suggests that it is less likely *Gm11549* RNA contributes to regulate mesendoderm differentiation.

Again, consistent with the EB differentiation functional impact from the NEMEP peptide frameshift mutation (*Nemep* KO), we found that mESCs harboring a promoter KO variant of the *Gm11549* locus (generated by CRISPR/Cas9) (Supplementary Fig. 4a and 5c) displayed defects during in vitro EB differentiation, including a sharply reduced extent of *Gm11549* induction and strong reductions in the expression of mesendoderm genes as shown by the transcriptome analysis (Fig. 3d). Interestingly, some pluripotency genes such as *Esrrb, Dnmt3l,* and *Utf1* were not

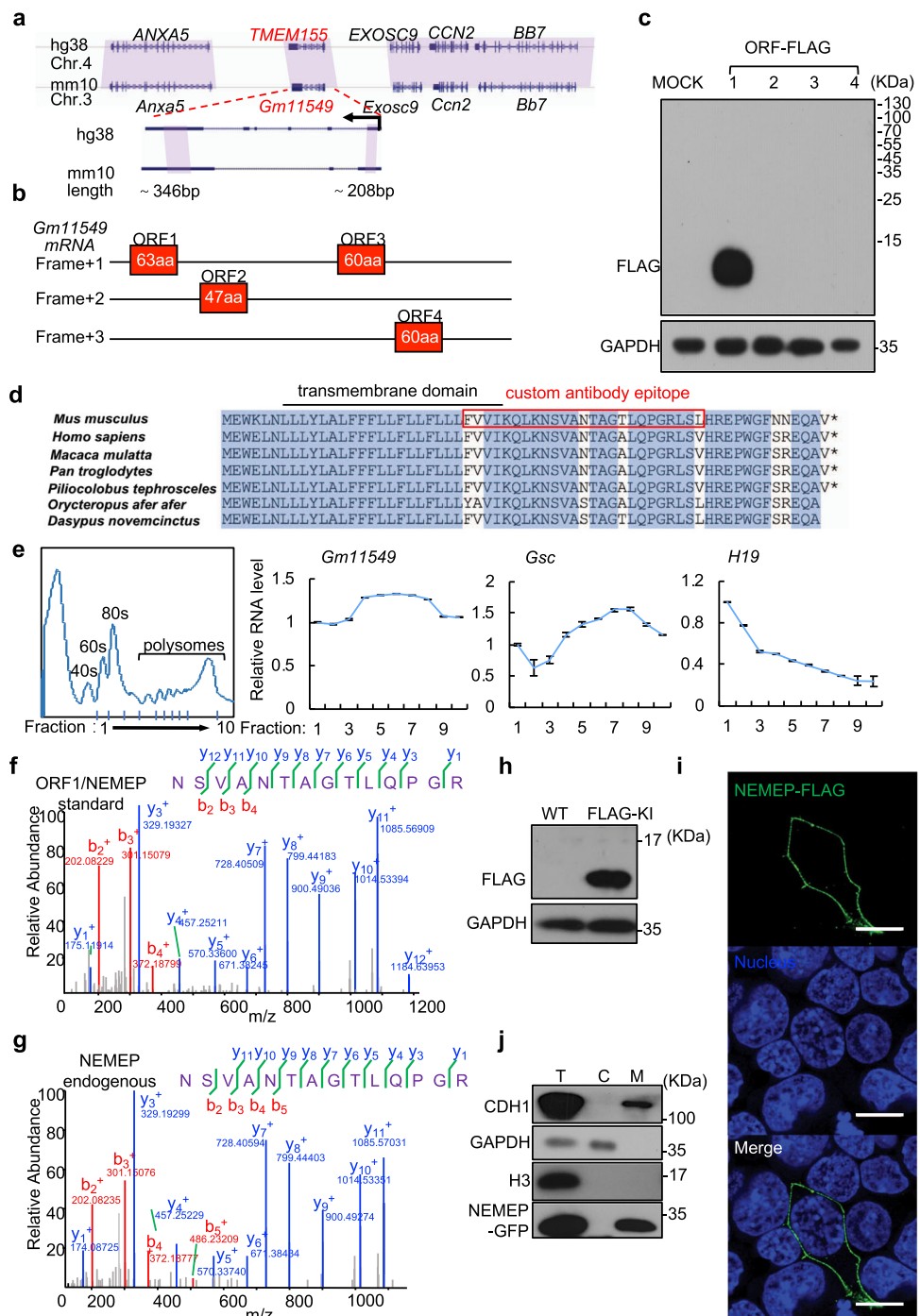

declined in *Gm11549* promoter KO EBs compared to the wild type EBs (Fig. 3d). Consistent with this notion of NEMEP-dependent progression towards a mesendoderm differentiation fate, the FOXA2, and T protein levels were significantly decreased in cells harboring mutant EBs including the frameshift and the promoter KO mutant cells by immunostaining assay (Fig. 3e, f). Offering further support, similar results were obtained in RNAi and CRISPRi-dCas9-KRAB experiments for depletion of *Gm11549* transcripts from EBs (Supplementary Fig. 5d, e). Thus, NEMEP is required for mesendoderm differentiation.

**NEMEP deletion leads to early defects in mouse embryo chimeras.** Embryonic chimeras of the mouse are well-established tools for studying cell lineage and cell potentials[48]. To further investigate the role of NEMEP in mesendoderm differentiation, we microinjected GFP-labeled WT, or *Nemep* KO (*Nemep*$^{-/-}$) mESCs into WT blastocysts to generate chimeric embryos, which were collected and examined at E7.5 (Fig. 4a). At E7.5, we observed no obvious differences between WT and *Nemep*$^{-/-}$ chimeras (Fig. 4b). Immunofluorescence analysis of the chimeras for expression of the mesendoderm marker FOXA2 and T showed that WT but not E7.5 *Nemep*$^{-/-}$ chimeras expressed FOXA2 and T (Fig. 4c, d). These chimeric embryo experiments strongly support that NEMEP is necessary for mesendoderm specification.

**NEMEP interacts with GLUT1/GLUT3 and promotes glucose uptake.** To explore the molecular mechanism of NEMEP in mesendoderm differentiation, we conducted pull-down experiments

**Fig. 2 Gm11549 encodes a transmembrane micropeptide: NEMEP. a** Conservation of the *Gm11549* and *TMEM155* loci between mouse (*Mus musculus*) and human (*Homo sapiens*). The purple shadow marks the conserved regions between mouse and human. The expanded region shows the conserved sequences between *Gm11549* and *TMEM155*. **b** Predicted ORFs in the *Gm11549* RNA sequence. **c** HEK293T cells were transiently transfected with expression vectors for all four of the predicted ORFs, each tagged with a FLAG epitope at their C termini. Immunoblotting analysis for accumulation of proteinaceous gene products from the predicted ORFs, detected with an anti-FLAG antibody. GAPDH was used as internal control. **d** Amino acid sequence alignment of ORF1 in the indicated species. **e** Ribosome profiling. Cytosolic lysates from differentiated cells were subjected to sucrose gradient centrifugation to isolate fractions including free 40/60 S subunits, monosomes, di/trisomes, and polysomes. RNAs were then extracted from these fractions and the *Gm11549*, *Gsc*, and *H19* levels were quantified by qPCR. *Gsc* served as the controls for coding transcripts and *H19* was the control for noncoding transcripts. **f** Following pull-down using FLAG antibody, a targeted proteomics analysis of lysates from transiently transfected HEK293T cells expressing C-terminal FLAG-tagged ORF1/NEMEP: MS/MS spectrum of one unique peptide corresponding to the *Gm11549* ORF1 protein (henceforth: "NEMEP" for Nodal Enhanced MEsendoderm microPeptide). **g** MS/MS spectrum for one unique peptide corresponding to NEMEP from a targeted proteomics analysis following pull-down using a NEMEP polyclonal antibody raised against a 25-residue region of the NEMEP from adult mouse brain tissue lysates. **h** Immunoblotting analysis of NEMEP-3xFLAG in differentiated WT and NEMEP-3xFLAG knock-in mESCs with anti-FLAG antibody. GAPDH was used as internal control. **i** Immunofluorescence detection of NEMEP-FLAG in NEMEP-FLAG expressing HEK293T cells with anti-FLAG antibody (green). Nuclei are stained with Hoechst33342 (blue). Original magnifications, 10 μm. **j** Subcellular localization of NEMEP-GFP in NEMEP-GFP overexpression mESCs. Total cell lysates (T), Cytosol (C), and membrane fractions (M) from NEMEP-GFP overexpression mESCs were assessed by immunoblotting against specific markers and GFP. The GAPDH, Histone H3, and CDH1 protein respectively served as markers for the cytosolic, nuclear, and membrane fractions. **c**, **e**, **f–j**: All data are representative of three independent experiments with similar results. Source data are provided as a Source Data file.

with EBs at day 3 using NEMEP-GFP as bait. Mass spectrometry analysis of co-purified proteins revealed two glucose transporters (GLUT1 and GLUT3) among the top-ranking candidate NEMEP-interacting proteins. Co-immunoprecipitation (co-IP) studies validated that both GLUT1 and GLUT3 do physically interact with NEMEP (Fig. 5a, Supplementary Data 1). Then, we confirmed that the homologous human NEMEP protein also interacts with the human GLUT1 and GLUT3 proteins (Fig. 5b). GLUT1 to GLUT4 are class I facilitative glucose transporters[49]. *Glut1* and *Glut3* have much higher expression in mESCs compared to *Glut2* or *Glut4* (Supplementary table 1). We conducted bimolecular fluorescence complementation (BiFC) analysis and successfully validated the interaction between NEMEP and these four of class I facilitative glucose transporters (Supplementary Fig. 6a).

Moreover, both confocal microscope images from BiFC assay and transient transfection of both GLUT1-FLAG or GLUT3-FLAG and NEMEP-HA vectors into mESCs show the colocalization of GLUT1 and GLUT3 with NEMEP on the cell membrane (Supplementary Fig. 6b, c). We also validated that NEMEP binds to endogenous GLUT1 and GLUT3 using NEMEP-3xFLAG knock-in mESCs (Fig. 5c). Further, deletion of the transmembrane domain (TMD) and of a 7-residue region (lacking H51-F57) of the NEMEP C terminus disrupted the interactions between NEMEP and GLUT1/GLUT3 proteins (Fig. 5d and Supplementary Fig. 6d). Domain swapping of the NEMEP-TMD domain for the ACVR1-TMD or ITGB1-TMD also impaired the interactions (Fig. 5d). These results demonstrate that the TMD domain and NEMEP residues H51-F57 are essential for NEMEP-GLUT1 and -GLUT3 interactions.

We next examined whether NEMEP impacts glucose transport by using the CRISPR-dCas9-VP64 activator system (CRISPRa) to generate mESCs that overexpress the endogenous mRNA encoding NEMEP (*Gm11549*) (Supplementary Fig. 7a). Compared to WT mESCs, we found that overexpression of NEMEP resulted in a ~30% increase in glucose consumption as well as a ~60% increases in lactate excretion (Fig. 5e, f). Consistently, a Seahorse-based analysis measuring the extracellular acidification rate (ECAR) showed that the glycolysis activity of the NEMEP-overexpressing mESCs was significantly higher than in WT mESCs (Fig. 5g and Supplementary Fig. 7b, c). Moreover, the overexpression of human NEMEP in HepG2 cells (Supplementary Fig. 7d) also significantly increased glucose consumption, lactate excretion, glycolysis activity (Supplementary Fig. 7e–g).

We then used Promega Glucose Uptake-Glo™ Assays to measure mESC glucose uptake to examine whether NEMEP impacts glucose transporter activity. Note that the measurement readout of the assay specifically reflects the glucose transporter activity. No changes in transporter activity were detected in mESCs expressing diverse NEMEP mutant variants (including TMD deletion, H51-F57 deletion, and the domain-swapped ACVR1-TMD and ITGB1-TMD variants); note that we did detect the expected increase in glucose transporter activity in mESCs expressing the full-length, wild type NEMEP (Fig. 5h).

Intriguingly, we detected NEMEP-mediated enhancements on both glucose uptake and glycolysis activities in mESCs in experiments examining overexpression of GLUT1 or GLUT3. Specifically, the expression of WT NEMEP but not TMD-deletion NEMEP or H51-F57-deletion NEMEP in mESCs overexpressing GLUT1 or GLUT3 resulted in significantly higher glucose uptake compared to the GLUT1 or GLUT3 overexpressing mESCs (Fig. 5i, j). Since H51-F57-deletion NEMEP failed to form complex with GLUT1 or GLUT3 (Fig. 5d), it suggests that NEMEP may act to facilitate glucose uptake through interaction with GLUT1 or GLUT3. However, it does not exclude the possibility that NEMEP might facilitate glucose uptake via other factors. In addition, the expression of WT NEMEP but not TMD-deletion NEMEP in mESCs overexpressing GLUT1 or GLUT3 resulted in significantly higher glycolysis activities compared to the GLUT1 or GLUT3 overexpressing mESCs (Supplementary Fig. 8a–h). Therefore, NEMEP's boosting glucose uptake likely through interactions with the known glucose transporters GLUT1 and GLUT3.

**NEMEP facilitates glucose uptake during mesendoderm differentiation.** We next explored the potential impacts of altered glucose uptake and differential glycolysis activity on mesendoderm differentiation. In vitro differentiation assays revealed that decreased glucose concentration in the culture medium reduces the expression levels of mesendoderm developmental marker genes in WT EBs (Fig. 6a). Further, knockdown of GLUT1 and GLUT3 led to severe defects in mesendoderm differentiation and in glucose uptake in EBs at day 3 (Fig. 6b, c). We also found that increasing the glucose concentration in the culture medium caused a significant reduction in the expression levels of multiple mesendoderm developmental marker genes in WT EBs (Fig. 6d). Thus, GLUT1 and GLUT3 are essential for normal mesendoderm

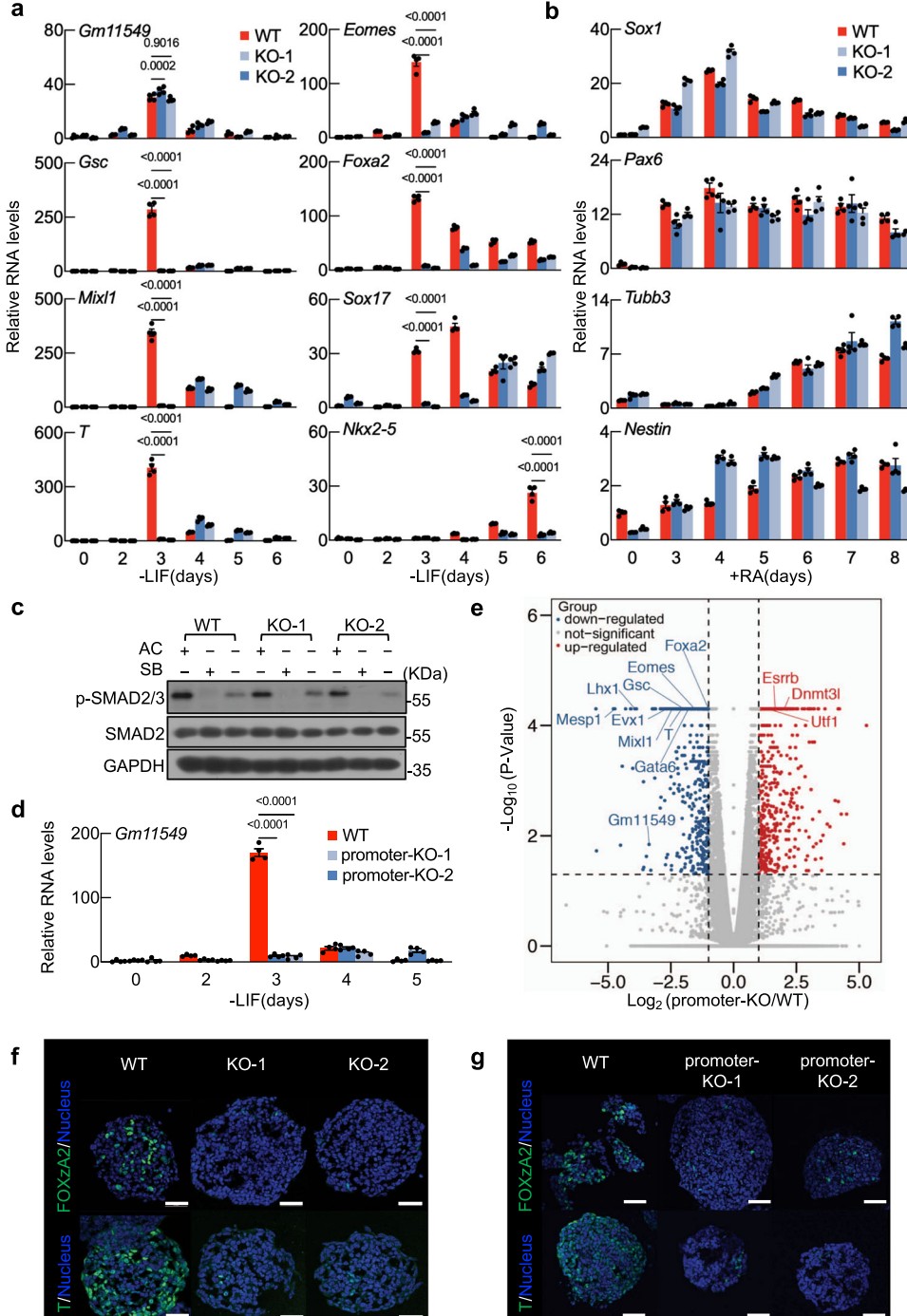

**Fig. 3 NEMEP is required for mesendoderm differentiation. a** NEMEP frameshift mutants (CRISPR/Cas9-edited frameshift of the exon1 region of the *Gm11549 locus*) (hereafter, *Nemep* KO) (Supplementary Fig. 3 a–c) and WT mESCs were induced for EB formation for the indicated durations. Total RNA was analyzed by qPCR using primers for the indicated genes. **b** *Nemep* KO and WT mESCs were induced for ectoderm differentiation for the indicated durations. Total RNA was analyzed by qPCR using primers for the indicated genes. **c** Immunoblotting analysis of indicated protein in WT and *Nemep* KO day 3 EBs treated with Activin A (AC) or SB431542 (SB) for 2 h. The data is representative of three independent experiments. **d** WT mESCs or mESCs with a promoter KO *Gm11549* locus were induced for EBs formation for the indicated durations. Total RNA was analyzed by qPCR using primers for *Gm11549*. **e** Volcano plot of transcriptome datasets of day 3 EBs of WT and *Gm11549* promoter KO cells (*n* = 2 biological independent samples). Blue, transcripts downregulated in *Gm11549* promoter KO versus WT EBs (fold change < 0.5, *p* < 0.05); Red, transcripts upregulated in *Gm11549* promoter KO versus WT EBs (fold change > 2, *p* < 0.05). **f** Immunofluorescence analysis of the mesendoderm marker FOXA2 and T in WT and *Nemep* KO EBs at day 3. The data are representative of three independent experiments. Scale bars, 50 μm. **g** Immunofluorescence analysis of the mesendoderm markers FOXA2 and T in WT and *Gm11549* promoter KO EBs at day 3. The data are representative of three independent experiments. Scale bars, 50 μm. **a**, **b**, and **d**: Data are the mean ± S.E.M, *n* = 4 biological independent samples. *P* values were determined by two-way ANOVA with Dunnett's corrections, and data are representative of three independent experiments with similar results. Source data are provided as a Source Data file.

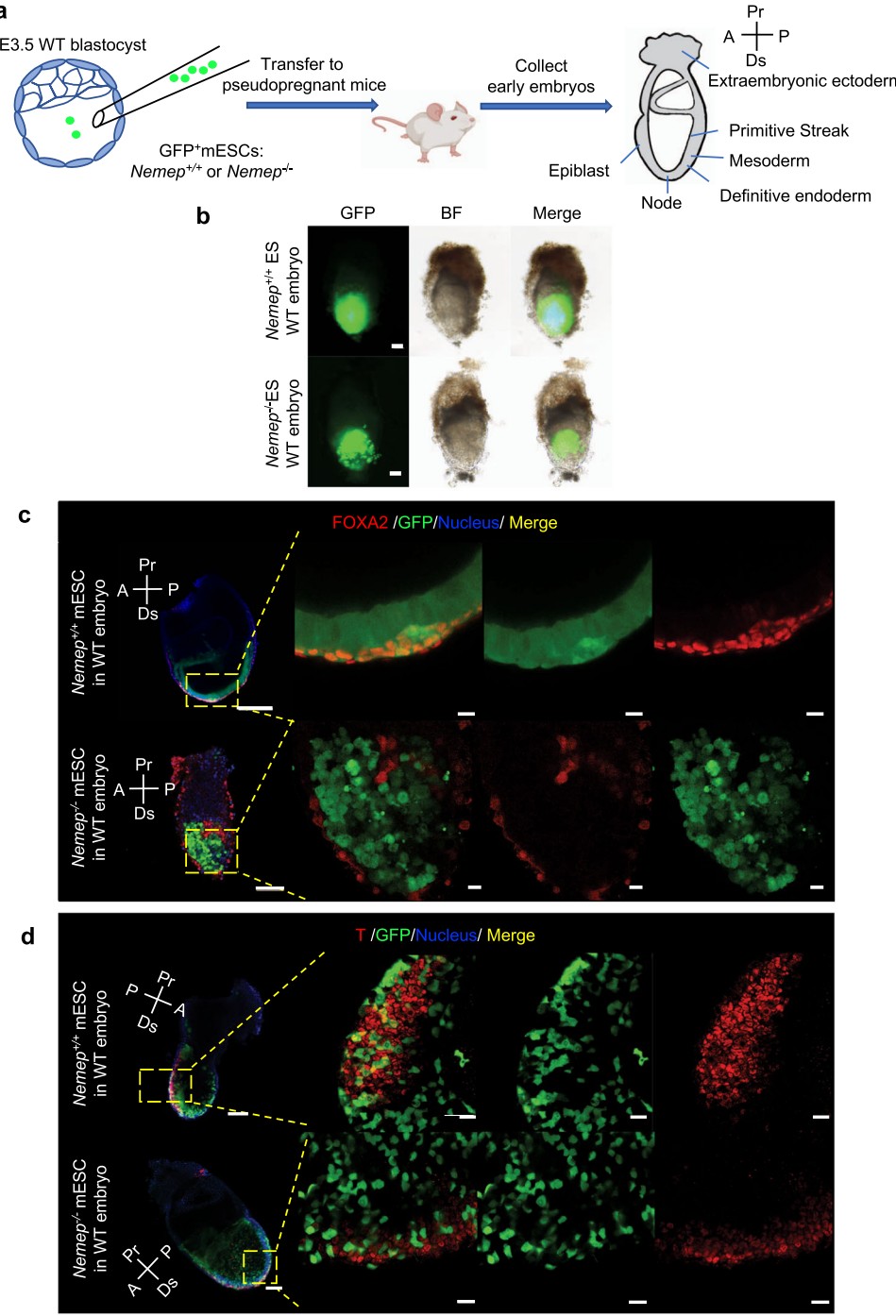

**Fig. 4 NEMEP deletion leads to early defects in mouse embryo chimeras. a** Scheme of blastocyst injection, transfer to pseudopregnant mouse, and collection of embryos at E7.5 (Created with BioRender.com). **b** Brightfield and green fluorescence (GFP) images of embryo chimeras including WT and *Nemep*−/− ESCs recovered at E7.5. Scale bars, 100 μm. Confocal microscopy images of serially sectioned embryo chimeras, with WT ESCs and *Nemep*−/− mESCs dissected at E7.5 depicting GFP (green), FOXA2 expression (red) (**c**) or T expression (red) (**d**), and nuclear (Hoechst33342, blue) localization. Scale bars, transverse section, 100 μm; high-magnification view, 20 μm. **b–d**: Data are representative of three independent experiments with similar results. Source data are provided as a Source Data file.

differentiation. Moreover, it appears that normal mesendoderm differentiation requires a developmentally-appropriate supply of glucose.

We then measured glucose uptake, ECAR, and oxygen consumption rate (OCR) in WT and *Nemep* KO EBs at day 3. Compared to WT EBs, the *Nemep* KO EBs displayed significant reductions in glucose uptake, glycolytic function, and mitochondrial respiration (Fig. 6e–j and Supplementary Fig. 9a–e).

Consistently, a metabolomics analysis detected significant declines in the levels of glycolysis and TCA cycle (Tricaboxylic Cycle) metabolites in *Nemep* KO EBs at day 3 (Fig. 6k). However, a companion RNA-seq analysis found no differences between day 3 WT and *Nemep* promoter KO EBs for the genes encoding glucose metabolism enzymes or glucose transporters (Supplementary Fig. 9f), and AKT activation which have been reported that regulates glucose uptake in other type of cells is not impacted

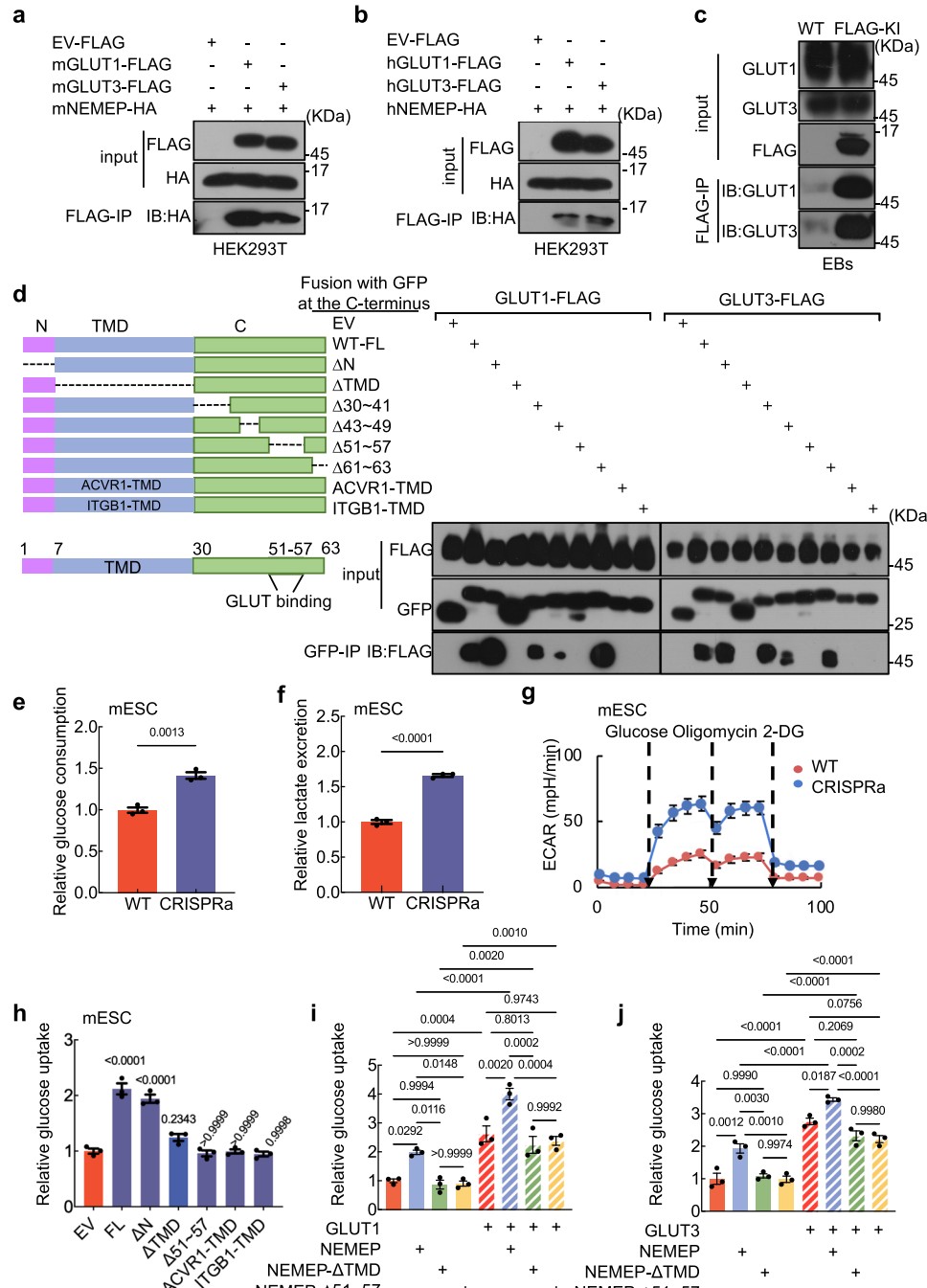

by depletion of NEMEP (Supplementary Fig. 9g)[50–52]. These results indicate that the aforementioned physical interaction of NEMEP and GLUT1/GLUT3 may support mesendoderm differentiation by facilitating glucose uptake.

Our analysis of data from a previously published transcriptome analysis of WT EBs from day 0 to day 4 revealed that a group of well-known glucose metabolism genes including *Glut1/Glut3* (*Slc2a1/ Slc2a3*) are up-regulated at day 3 EBs[44], the same time point that mesendoderm genes are induced (Supplementary Fig. 9h).

Despite previous reports that TGF-β signaling can impact glucose metabolism in mesangial cells and NIH 3T3 cells[53–55], we are unaware of any studies examining whether Nodal signaling regulates glucose metabolism during mesendoderm differentiation. Pursuing this, we found that treatment of day 3 EBs with the known Nodal signaling activator Activin significantly increased glucose uptake (Fig. 7a). Further, compared to WT EBs the extent of glucose uptake was significantly reduced in EBs deficient for the Cripto and Cryptic, deficient for SMAD2 and SMAD3, and deficient for TRIM33 (Fig. 7b–d). Moreover, Activin treatment does not induce glucose uptake in EBs deficient for SMAD2 and SMAD3, or deficient for TRIM33 compared to WT EBs (Fig. 7c, d). Noteworthy, Activin treatment induces glucose uptake in EBs deficient for the Cripto and Cryptic (Fig. 7b), since Cripto/ Cryptic-depleted cells remained responsive to Activin[43]. Together, these in vitro findings indicate that the Nodal signaling functions to positively regulate glucose uptake during mesendoderm differentiation. However, given that the expression of well-known glucose metabolism genes (including *Slc2a1* and *Slc2a3*) was not altered by activation (Activin A treatment) or inhibition (SB431542 treatment) of Nodal signaling activity (Supplementary Fig. 9i), we suspected that the Nodal-signaling-mediated

**Fig. 5 NEMEP interacts with GLUT1/GLUT3 and facilitates glucose uptake.** Physical interactions of mouse (**a**) human (**b**) NEMEP with GLUT1 and GLUT3. Lysates from HEK293T cells co-transfected with plasmids encoding GLUT1-FLAG or GLUT3-FLAG or control vector and NEMEP-HA (as indicated) were immunoprecipitated with anti-FLAG affinity beads, and immune complexes were analyzed by immunoblotting using an antibody against HA. The protein inputs were detected with western blotting using indicated antibodies. **c** Immunoprecipitation of endogenous NEMEP from NEMEP-3XFLAG knock-in mESCs using anti-FLAG antibody. The immune complexes were analyzed by immunoblotting using antibodies against FLAG and GLUT1 and GLUT3. The protein inputs were detected with western blotting using indicated antibodies. **d** Lysates from HEK293T cells co-transfected with plasmids encoding mGLUT1-FLAG or mGLUT3-FLAG and NEMEP (GFP-tagged, wild type or indicated mutant variants) immunoprecipitated with anti-GFP-trap affinity beads; immune complexes were analyzed by immunoblotting using an antibody against FLAG. The protein inputs were detected with western blotting using indicated antibodies. Glucose consumption (**e**) and lactate excretion (**f**) in WT and CRISPR-dCas9-VP64 (CRISPRa) mediated NEMEP overexpressing mESCs. Data are means ± S.E.M., $n = 3$ biological independent samples. **g** WT and CRISPR-dCas9-VP64 (CRISPRa) mediated NEMEP overexpressing mESCs were supplied with 25 mM glucose, 2 µM of oligomycin (ATP synthase inhibitor), and 50 mM 2-DG (a glucose analog that inhibit glycolysis) at the indicated time. ECAR was examined using a Seahorse XFe96 analyzer. The values are normalized to the protein concentration (means ± S.E.M., $n = 8$ biological independent samples). **h** Glucose uptake analysis in mESCs cells stably expressing control plasmid or plasmids with WT or mutant variants of NEMEP, as indicated. The values are normalized to the protein concentration (means ± S.E.M., $n = 3$ biological independent samples). **i,j** Glucose uptake analysis in HEK293T cells expressing plasmids for overexpression of the indicated proteins or protein pairs. The values are normalized to the protein concentration (means ± S.E.M., $n = 3$ biological independent samples). **a–j**: Data are the representative of three independent experiments with similar results. $P$ values were determined by unpaired two-tailed $t$-test (**e**, **f**), one-way ANOVA (**h**) with Sidak's corrections or two-way ANOVA (**i**, **j**) with Tukey's corrections. Source data are provided as a Source Data file.

promotion of glucose uptake is through unknown glucose metabolism genes, and NEMEP may be one of them.

To explore potential mechanisms that may reconcile our observation of Nodal signaling's promotion of glucose uptake during mesendoderm differentiation with our findings that the NEMEP-GLUT1/GLUT3 physical interaction and NEMEP promoting glucose uptake, we conducted experiments combining stimulation of Nodal signaling and genetic knockout of NEMEP. Consistently, we found that the Activin-induced increase in glucose uptake was significantly stronger in WT EBs than in *Nemep* KO EBs (Fig. 7e). An expression of NEMEP in the *Nemep* KO EBs rescued the defect of glucose uptake (Fig. 7f and Supplementary Fig. 9j). Given that NEMEP KO does not affect Nodal signaling activity (Fig. 3c) and the fold induction in response to Activin remains the same as in control cells for glucose uptake (Fig. 7e), we suspect that: (i) Nodal signaling induces the transcriptional activation of other, as-yet-unknown genes that somehow regulate glucose uptake; (ii) crosstalk of Nodal signaling with other signaling pathways to facilitate glucose uptake; (iii) Nodal signaling may modulate post-translational modifications (i.e. phosphorylation) of proteins involved in facilitating glucose uptake[56]. Hence, even if NEMEP is missing, the cells can respond to Nodal signaling to regulate glucose uptake, but at relatively lower level.

That is, our results support a model of mesendoderm differentiation from pluripotent mESCs wherein activated Nodal signaling induces the expression of a micropeptide, NEMEP that may function by interacting with GLUT1 and GLUT3 or other facilitators involved in glucose uptake, and augmenting the glucose uptake to meet the energy needs during mesendoderm differentiation (Fig. 7g).

## Discussion

Our study identified a highly conserved cytosolic lncRNA *Gm11549* as a direct target of Nodal signaling during mesendoderm differentiation. Interestingly, there is an "ORF" hidden within *Gm11549* that encodes a transmembrane micropeptide which we named NEMEP. We discovered that NEMEP is essential for mesendoderm differentiation in mESCs, a finding that both expands the list of Nodal target genes essential for mesendoderm differentiation and provides a clear example for a functional micropeptide acting as a developmental "stage-specific facilitator" during early development. Given the widespread temporal and spatial expression patterns for many lncRNAs

during early development[46], it seems likely that additional "lncRNA-coding peptides" will be identified along with improvements in bioinformatics and biological research technologies[57–61]. It is conceivable that such a "facilitator" role for micropeptides could represent a common mechanism for achieving the required fine-scale temporal and spatial regulation to support proper cell fates in early development.

Studies based on ribosomal profiling and mass spectrometry showed that a fraction of putative small open reading frames (sORFs) hidden in some cytosol-localized lncRNAs are translated into functional micropeptides[62–64]. These micropeptides (shorter than 100 amino acids) are essential in diverse biological processes[62,65–71]. We speculate that there could be more micropeptides regulated by TGF-β members in different cell contexts. Experimentally deciphering the in vivo production of such micropeptides and their molecular functions can deepen our understanding of the apparently quite broad and mechanistically diverse impacts of TGF-β signaling during early development.

Currently, studies of micropeptides are still in their infancy, and there is insufficient data to enable the definition of a canonical mode of action for these biomolecules. Nevertheless, there are examples for a group of micropeptides that function through their interactions with larger proteins[64]. By interacting with larger proteins, these micropeptides are involved in regulating the activities of protein complex machinery. Our work in the present study demonstrates that NEMEP interacts with the GLUT1/GLUT3 glucose transporter proteins and augments glucose uptake during mesendoderm differentiation. We validated that glucose is essential for mesendoderm differentiation, and demonstrated that loss of NEMEP both significantly impaired mesendoderm differentiation and decreased glucose uptake. Intriguingly, Nodal signaling induces glucose uptake through NEMEP, and NEMEP can rescue the defect of glucose uptake in *Nemep* KO cells. Understood together, these results establish that the Nodal signaling mediated induction of NEMEP regulates mesendoderm differentiation by facilitating glucose uptake.

The precise biochemical mechanism whereby NEMEP cooperates with GLUT1/GLUT3 to facilitate glucose uptake remains to be elucidated. Although we have lines of evidence supporting the interaction of NEMEP and GLUT1/GLUT3 (BiFC, co-IP), the precise nature of their binding interactions is not yet clear. If the interaction is direct, then it will be interesting to solve a crystal or cryo-EM structure of NEMEP in complex with GLUT1/GLUT3, which should help reveal how the interaction with the glucose transporters facilitates glucose uptake. If the interaction is

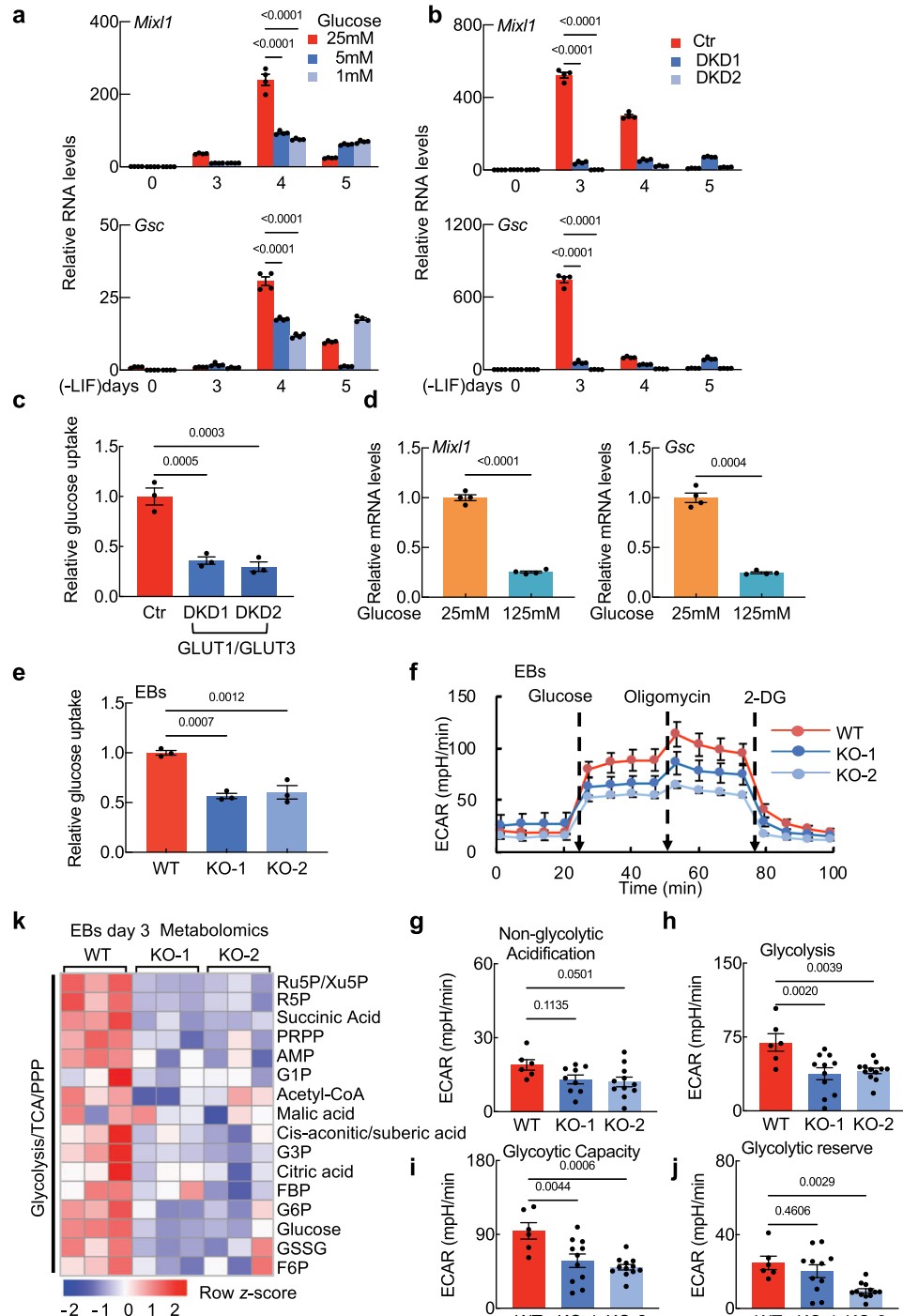

**Fig. 6 NEMEP facilitates glucose uptake during mesendoderm differentiation. a** mESCs cultures were induced for EBs formation for the indicated lengths of time. EBs at day 2 were cultured in medium containing 25 mM, 5 mM, or 1 mM glucose. Expression of the indicated genes was analyzed by qPCR ($n = 4$ biological independent samples). **b** qPCR analysis of indicated genes at the times in EBs induced from WT or *Glut1 and Glut3* double knock-down (DKD) cells by shRNA ($n = 4$ biological independent samples). **c** Glucose uptake analysis in WT or *Glut1* (*Slc2a1*) *and Glut3* (*Slc2a3*) double knock-down (DKD) EBs. The values are normalized to the protein concentration ($n = 3$ biological independent samples). **d** EBs at day 2 were cultured in medium containing 125 mM, 25 mM, or 1 mM glucose. Expression of the indicated genes was analyzed by qPCR at EBs day 3 ($n = 4$ biological independent samples). **e** Glucose uptake analysis in WT and NEMEP KO EBs. The values are normalized to the protein concentration ($n = 3$ independent samples). **f–j** WT and *Nemep* KO EBs at day 3 were supplied with 25 mM glucose, 2 μM oligomycin, and 50 mM 2-DG at the indicated times. ECAR was examined using Seahorse XFe96 analyzer (**f**). Normalized to the protein concentration ($n = 8$ biological independent samples). Relative non-glycolytic acidification (**g**), glycolysis levels (**h**), glycolytic capacity (**I**), and glycolytic reverse (**j**) were normalized to the protein concentration (at least 6 biological independent samples per genotype). **k** Heatmap displaying glycolysis, TCA cycle, and pentpentose phosphate pathway metabolites in WT and *Nemep* KO EBs at day 3; these data are from a targeted metabolomics profiling analysis ($n = 3$ biological independent samples). **a–j**: Data are the mean ± S.E.M. *P* values were determined by unpaired two-tailed *t*-tests (**d**), one-way ANOVA (**c**, **e–j**) with Dunnett's corrections or two-way ANOVA (**a**, **b**) with Dunnett's corrections, and data are representative of three independent experiments with similar results. Source data are provided as a Source Data file.

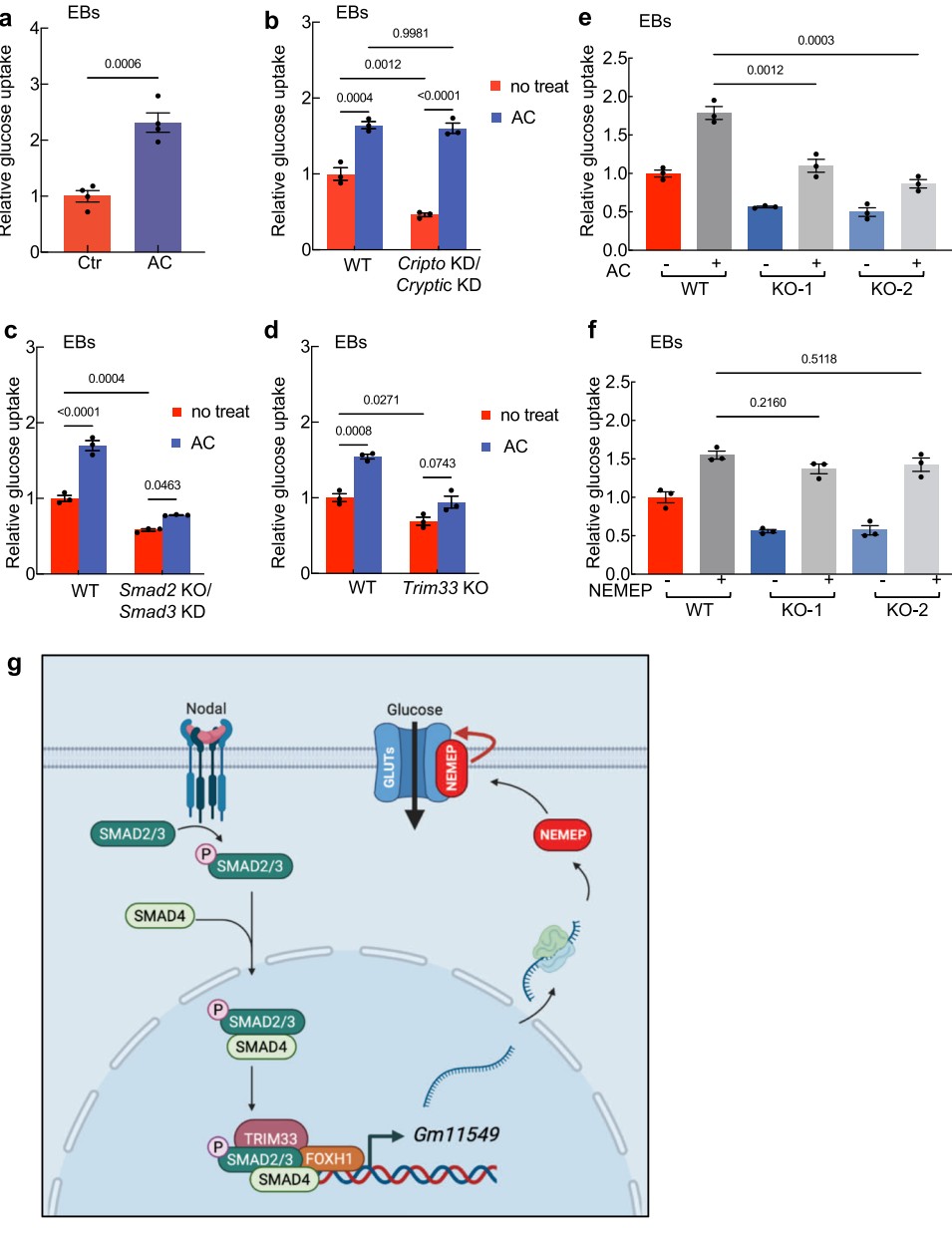

**Fig. 7 Nodal signaling mediates glucose uptake during mesendoderm differentiation. a** Glucose uptake analysis in control and Activin A (AC)-treated for 2 h EBs. The values are normalized to the protein concentration ($n = 4$ biological independent samples). **b** Glucose uptake analysis in Activin A (AC)-treated for 2 h or untreated WT and *Cripto* KD/*Cryptic* KD EBs. The values are normalized to the protein concentration ($n = 3$ biological independent samples). **c** Glucose uptake analysis in Activin A (AC)-treated for 2 h or untreated WT and *Smad2* KO/*Smad3* KD EBs. The values are normalized to the protein concentration ($n = 3$ biological independent samples). **d** Glucose uptake analysis in Activin A (AC- treated for 2 h or untreated WT and *Trim33* KO EBs). The values are normalized to the protein concentration ($n = 3$ biological independent samples). **e** Glucose uptake analysis of Activin A (AC)-treated for 2 h or untreated WT and *Nemep* KO EBs at day 3. The values are normalized to the protein concentration ($n = 3$ biological independent samples). **f** Glucose uptake analysis of NEMEP-FLAG overexpressing WT and *Nemep* KO EBs at day 3. The values are normalized to the protein concentration ($n = 3$ biological independent samples). **g** Working model for the role of NEMEP during mesendoderm differentiation (Created with BioRender.com). A highly conserved putative lncRNA *Gm11549* is specifically induced by Nodal signaling during mesendoderm differentiation and it is essential for mesendoderm differentiation in mESCs. There is a "ORF" hidden in *Gm11549* that codes a transmembrane micropeptide, we named NEMEP. NEMEP functions by interacting with GLUT1 and GLUT3, and augment the glucose uptake capacity of glucose transporter proteins to meet the energy needs during mesendoderm differentiation. **a**-**f**: Data are the mean ± S.E.M. *P* values were determined by unpaired two-tailed *t*-tests (**a**), one-way ANOVA (**e**, **f**) with Dunnett's corrections, two-way ANOVA (**b**-**d**) with Sidak's corrections, and data are representative of three independent experiments with similar results. Source data are provided as a Source Data file.

indirect, then identification of other components in the complex should still enable similar structure-based studies to reveal NEMEP's glucose-uptake augmentation mechanism.

Beyond its long-understood central functions in cellular energy production, there is growing evidence supporting that glycolysis must be understood as a crucial regulator of cell identity under physiological conditions[4,5,72]. For example, a study of human pluripotent stem cells demonstrated metabolic switching from glycolysis to oxidative phosphorylation during mesoderm and endoderm differentiation; this switch was not detected during

ectoderm differentiation[3]. A seminal genetics study demonstrated that glycolysis is necessary for gastrulation in mouse[5]. The primitive streak is the organizing center for amniote gastrulation; this region is known to define the future embryo midline and to serve as a conduit of cell migration for germ layer formation[73,74]. The RNA transcript for NEMEP is specifically expressed in the primitive streak of E7.0 mouse embryos. And the expression pattern is highly similar to that of mesendoderm LDTFs (e.g., *Mixl1*, *Gsc*, *T*, and *Foxa2*) which are Nodal signaling transcriptional target genes. Given its function in augmenting glucose uptake, it is plausible to speculate that the induction of NEMEP by Nodal signaling at the primitive streak "turns on" a precisely spatiotemporally regulated additional glucose source to meet the highly specific energy needs of a small number of cells at this development stage. At minimum, our discoveries provide another line of evidence illustrating how metabolism actively regulates cell identity during embryonic development.

Our study also has implications outside of early development. Recall our finding that the brain is apparently the only adult tissue in which NEMEP is highly expressed. Given the known role of glucose as the required fuel for brain function under non-starvation conditions (more than 40% of the blood glucose is consumed by brain)[75,76], we speculate that NEMEP may function to help meet the energy requirements for brain function. Conditional knockout mice will be helpful for elucidating this potential role of NEMEP in the brain, as will profiling of NEMEP expression in particular brain structures in differential states of activation. It is conceivable that specific cells in the brain may employ NEMEP to augment glucose uptake in a manner similar to our observations from mesendoderm differentiation.

## Methods

**Animal ethics statement**. Experiments involving mice were all in accordance with institutional guidelines for the care and use of animals. The animal experiments that were conducted as part of this research were completed in accordance with the guidelines provided by the Tsinghua University Institutional Animal Care and Use Committee (IACUC) and were in compliance with the relevant ethical regulations regarding animal research. All animal procedures were approved by Institutional Animal Care and Use Committee of Tsinghua University (IACUC:16-XQR2). C57BL/6J mouse strains were obtained from Laboratory Animal Research Center in Tsinghua University. C57BL/6J mouse strains were housed under SPF (specific-pathogen-free) at 20–22 degree with 12 h:12 h light: dark cycles at 50–60% humidity.

**Generation of chimeric embryos**. GFP-expressing WT and *Nemep*$^{-/-}$ (*Nemep* KO) mESCs were micro-injected after culturing for 3 days on irradiated mouse embryonic fibroblast (MEF) feeder layers. 15 to 20 mESCs from each group were injected into embryonic day (E) 3.5 blastocysts (C57BL/6J). Injected blastocysts were cultured in KSOM/AA (Millipore) at 37 °C in an atmosphere of 5% $CO_2$ for 2–3 h to allow for recovery of blastocyst morphology and then implanted into the uterine horns (up to 10 embryos per horn) of E2.5 pseudopregnant females (C57BL/6J). Chimeric embryos were recovered at E7.5. All the C57BL/6J mouse strains were obtained from Laboratory Animal Research Center in Tsinghua University. Female mice were used between ages 6–8 weeks.

**Cell lines**. Mouse ES cell lines E14Tg2a.IV were purchased from the American Type Culture Collection (ATCC; https://www.atcc.org:443/). mESCs E14Tg2a.IV, CCE, *Smad4* null (BNN)[77], *Smad2* null (KT15)[78], *Smad3* KD/*Smad2* null, and *Trim33* null[25] lines were maintained on 0.1% gelatin-coated plates in LIF-supplemented medium at 37 °C with 5% $CO_2$ as previously described[25,79]. Basic mESC medium contains Dulbecco's modified Eagle's medium (DMEM, Thermofisher), 15% Fetal Bovine Serum (FBS) (ExCell Bio), 1% penicillin-streptomycin-amphotericin B solution (Biological Industries), 1% non-essential amino acids (Thermofisher), 1% L-glutamine (Biological Industries), 1% sodium pyruvate (Sigma), 100 μM β-mercaptoethanol (Sigma), 103 U/mL mouse LIF. HEK293T and Hep G2 cells were cultured in DMEM supplemented with 10% FBS (ExCell Bio) and 1% penicillin-streptomycin-amphotericin B solution. We maintained all cell lines at 37 °C in a 5% $CO_2$ cell culture incubator and tested all cell lines routinely for mycoplasma contamination.

**mESCs differentiation**. Mouse EBs formation and differentiation were carried out as described by the supplier (ATCC). Briefly, for mesoderm differentiatrion,

mESCs were cultured in mESC medium without LIF in an ultra-low attachment dish, embryoid bodies(EBs) were collected at the indicated time points.Ectoderm differentiation assays were carried out as described previously[80,81]. Briefly, mESCs were cultured in mESC medium without LIF in an ultra-low attachment dish, and 1 μM Retinoic acid was added in culture medium. Samples were collected at the indicated time points, and medium was changed every other day.

**shRNA mediated transcript knocking down**. Annealed shRNAs oligonucleotides were cloned into AgeI- and EcoRI-digested pLKO.1-puro lentiviral vector (Addgene #8453). shRNA sequences for *Gm11549*, *Slc2a1*, and *Slc2a3* are listed in Supplementary Table 2. Lentivirus packaging was performed as previously described[25]. HEK293T cells were plated onto a 10 cm dish, and then were transfected with 12 μg pLKO.1 vector along with 6 μg pMDL-RRE(packaging plasmid), 3 μg pVSVG (envelope plasmid), and 3 μg pRev (packaging plasmid) and using Lipofectamine 2000 reagent (Invitrogen), following the manufacturer's protocol. Medium was changed 24 h later. The lentivirus-containing medium were collected at 48 h and 72 h post-transfection and incubated overnight at 4 °C with lenti-X concentrator (631232, TaKaRa), and were then spun at 3000 g at 4 °C for 30 min. The viral pellets were used for transducing the cells.

**Rapid amplification of cDNA ends (RACE)**. 5′ and 3′ RACE was performed using the primers listed in Supplementary Table 3 and the SMART RACE cDNA Amplification Kit (Clontech), according to the manufacturer's recommendations.

**CRISPR-Cas9 mediated genome editing**. The designated sgRNA sequences were designed using online software (https://portals.broadinstitute.org/gpp/public/analysis-tools/sgrna-design).

The annealed oligonucleotide pairs for the mutants: *Gm11549* promoter knock-out (KO) (promoter-KO-1 and promoter-KO-2), exon KOs (exon-1 KO, exon-2 KO, exon-3 KO-1, and exon-3 KO-2), NEMEP frameshift mutants (KO-1 and KO-2) and 3xFLAG knock-in (KI), was ligated into BbsI-linearized pSpCas9 (BB)-2A-GFP (PX458) (Addgene #48138)[82].

To generate *Gm11549* promoter KOs and variant exon KOs mESCs, two PX458-sgRNAs to designated mutants were co-transfected into mESCs using Lipofectamine 2000 (Life Technologies) according to the manufacturer's instructions. 48 h post-transfection, GFP-positive cells were sorted by Fluorescence-activated Cell Sorting (FACS) as a single cell and cultured in a 96-well plate for 2 weeks. PCR-based genotyping was applied to pick clones showing the deletion of the targeted region in Gm11549/NEMEP genomic DNA.

To generate NEMEP frameshift mutants (*Nemep* KO) mESCs, PX458-sgRNA was transfected into mESCs using Lipofectamine 2000. 48 h post-transfection, GFP-positive cells were sorted by FACS as a single cell and cultured in a 96-well plate for 2 weeks. PCR primers flanking the targeted region were used to amplify the affected region for genotyping. After AfeI (NEB) digestion for 2 h at 37 °C, the PCR product was electrophoresed in an agarose gel to identify homozygous wild-type, homozygous knockout, and heterozygous clones.

To generate NEMEP-3xFLAG KI mESCs, PX458-sgRNA and the donor plasmid were co-transfected into mESCs using Lipofectamine 2000. 48 h post-transfection, GFP-positive cells were sorted by FACS sorted as a single cell and cultured in a 96-well plate for 2 weeks. PCR primers flanking the homologous arms were used. After EcoRV (NEB) digestion for 2 h at 37 °C, the PCR products were electrophoresed in agarose gel to identify wild-type and 3xFLAG knock-in clones, which were confirmed by immunoblot.

For activating endogenous Gm11549 expression (CRISPRa)[83]: sgRNAs were cloned into MS2-sgRNA-Zeo (Addgene #61427) using the BsmBI restriction sites. Lentiviruses containing each of the plasmids (dCAS-VP64-Blast (Addgene #61425), MS2-P65-HSF1-Hygro (Addgene #61426), and MS2-sgRNA-Zeo) were generated, and mESCs were infected with a 1:1:1 mix of the viruses in the presence of 8 μg/mL polybrene (Sigma). Infected cells were selected for 48 h with 10 μg/mL blasticidin (Selleck), 250 μg/mL hygromycin (BIOBYING), and 250 μg/mL zeocin (InvivoGen).

For repressing endogenous Gm11549 expression (CRISPRi)[84]: sgRNAs were cloned into MS2-sgRNA-Zeo (Addgene #61427) using BsmBI restriction sites. Lentiviruses containing each of the plasmids (dCAS9-KRAB-Blast (Addgene #50919) and MS2-sgRNA-Zeo) were generated, and mESCs were infected with a 1:1 mix of the viruses in the presence of 8 μg/mL Polybrene. Infected cells were selected for 48 h with 10 μg/mL blasticidin and 250 μg/mL zeocin. All sgRNAs and PCR primers sequences are listed in Supplementary Tables 4 and 5.

**RNA-seq and ChIP-seq**. The RNA-seq library preparation and sequencing were performed as described[44]. For RNA-seq data analysis, we first evaluated RNA-seq reads quality using FastQC (version 0.10.1). Then, the reads were mapped to the mouse reference genome (mm10) using TopHat (version 2.0.10) with default parameters, and the mouse reference genome sequence was downloaded from Ensembl (Mus musculus GRCm38/mm10). Next, the gene expression levels (Reads Per Kilobase per Million mapped reads; RPKM) of annotation genes were estimated using Cufflinks (version 2.2.1) by providing the mouse genome annotation from GENCODE (version M4). Differential testing and log2 fold change

calculations were performed using Cuffdiff (version 2.2.1), with the implementation of two biological replicates.

We analyzed the ChIP-seq data (TRIM33, SMAD2/3, SMAD4, FOXH1) as previously described[21,44]. Briefly, the data were trimmed using TrimGalore (version 0.6.1) and then aligned to mm10 using Bowtie2 (version 2.3.3). We chose MACS2 (version 2.1.4) to call the peaks using the "broad peaks" setting for SMAD2/3, SAMD4, TRIM33, and FOXH1. We subsequently generated bigwig files from the bam files using the Coverage function in deepTools (version 3.4.4). For visualization purposes, we normalized the data to 1X genome coverage (mm10). Representative track diagrams were generated using the Integrated Genomics Viewer software (version 2.8.13).

**Subcellular fractionation assay**. For RNA subcellular fractionation, EBs and monolayer differentiation samples were harvested. Nucleus and cytoplasm RNA fractions were extracted as previously described[85]. Cells were harvested and washed by 10 mL ice-cold PBS, then spun at 700 g for 3 min. The cell pellets were resuspended in 200 μL cold cytoplasmic lysis buffer (0.15 NP-40, 10 mM Tris pH 7.5, 150 mM NaCl), and incubated on ice for 5 min. The lysates were transferred onto 500 μL ice-cold sucrose buffer (10 mM Tris pH 7.5, 150 mM NaCl, 24% sucrose), spun at 12,000 g for 10 min. The supernatants were collected as the cytoplasmic fraction. The pellets were collected and used for nuclear RNA extraction. Ratios of interesting genes in different subcellular fractions were calculated by normalizing the expression levels of quantitative RT-PCR to the relative volume of each fraction used for RNA analysis. For protein subcellular fractionation, we used a previously published protocol (https://bio-protocol.org/e754).

**Immunofluorescence**. For the attached cells, samples were fixed in 4% PFA in PBS for 30 min at room temperature. The cells were permeabilized with 0.5% Triton X-100 in PBS (PBST) for 10 min at room temperature and blocked in 1% BSA in PBS for 1 h at room temperature. Then, the samples were incubated with primary antibodies in 1% BSA in PBST overnight at 4 °C. The next day, the coverslips were incubated with DyLight594-Goat Anti-Mouse IgG (H + L) (Huaxingbio,1:1000) for 1 h at room temperature. Then, samples were incubated with Hoechst 33342 (Thermo Fisher Scientific) for 5 min at room temperature followed by three PBS washes.

For EBs, EBs at day 3.5 were fixed in 4% PFA in PBS for 30 min at room temperature, incubated in a solution of 30% sucrose in PBS for 1–2 days, and processed for OCT embedding. EB sections were permeabilized for 30 min in 0.5% PBST, followed by 1% BSA in PBST for blocking. Then, samples were incubated overnight with the primary antibody for FOXA2 (Cell Signaling Technology, 8186) or T/Brachyury (Abcam, 209665) at a dilution of 1:200 at 4 °C, washed three times with PBS, followed by a 1 h incubation with IF 488-Goat Anti-Rabbit IgG (H + L) (Huaxingbio) or DyLight 488-Donkey Anti-Goat IgG (H + L) (Huaxingbio) at a dilution of 1:1000 in 1% BSA in PBST. Finally, samples were incubated with Hoechst 33342 for 5 min in room temperature followed by PBS washing three times.

For E7.5 embryo wholemount immunostaining, after dissection of E7.5 embryos, embryos were imaged wholemount using a Nikon fluorescence stereomicroscope to assess the level of chimerism based on GFP fluorescence prior to fixation. Embryos were fixed for 15 min at room temperature in 4% PFA (Electron Microscopy Sciences) then washed with PBS with 0.1% Triton X-100. Embryos were permeabilized with 0.5% PBST for 30 min at room temperature and blocked overnight at 4 °C in blocking buffer (0.1% PBST) with 5% horse serum (Solarbio) and 1% BSA. The next day, embryos were transferred to primary antibodies diluted in blocking buffer and incubated overnight at 4 °C. Primary antibodies were used at the following dilutions: T/Brachyury (1:200) and FOXA2 (1:200). The next day, embryos were washed twice in PBS with 0.1% Triton X-100 for 10 min at room temperature and blocked for approximately 6 h in blocking buffer followed by overnight incubation in blocking buffer with secondary antibodies (1:500) at 4 °C. Embryos were then washed three times for 10 min in PBS with 0.1% Triton X-100 at room temperature. The final wash contained 5 μg/mL Hoechst 33342.

For imaging, embryos were positioned in glass-bottom dishes (NEST) in PBS and imaged using a Nikon A1R HD25 confocal microscope equipped with 405 nm, 488 nm, and 594 nm solid lasers and analyzed by NIS-Elements Viewer 4.11.0 and Image J V1.50e. A 60x/100x oil immersion objective lens was used for captured images.

**RNA fluorescence in situ hybridization (RNA FISH)**. mESCs were set up to differentiation for 3 days and grown on coverslips in 12-well plates, briefly washed with PBS, and fixed with PBS/3.7% formaldehyde at room temperature for 10 min. Following fixation, cells were washed twice with PBS. The cells were then permeabilized in 70% ethanol for at least 1 h at 4 °C. Stored cells were briefly rehydrated with Wash Buffer (2 × SSC, 10% formamide) (Biosearch) before FISH. The Stellaris FISH Probes (Gm11549, Gapdh, and Malat1) were added to the hybridization buffer (2 × SSC, 10% formamide, 10% dextran sulfate) (Biosearch) at a final concentration of 250 nM. Hybridization was carried out in a humidified chamber at 37 °C overnight. The following day, the cells were washed twice with Wash Buffer at 37 °C for 30 min each. The third wash contained 5 μg/mL Hoechst33342 for nuclear staining.

**Extracellular acidification rate (ECAR) and oxygen consumption rate (OCR)**. ECAR and OCR were analyzed on an XF96 Extracellular Flux Analyzer (Agilent) according to the manufacturer's recommendations. For Seahorse flux analysis, cells were plated in XF 96-well microplates (Agilent). After 24 h, the medium was completely replaced with XF Base medium (Aligent) (ECAR: XF Base Medium containing 4 mM L-glutamine. OCR: XF Base Medium containing 25 mM glucose (Sigma), 4 mM L-glutamine and 2 mM sodium pyruvate). Reagents were added at the following final concentrations: for ECAR measurement, glucose (25 mM), oligomycin (2 μM) (Abcam), 2-DG (100 mM). For OCR measurement, FCCP (1 or 2 μM) (Sigma), rotenone (1 μM) (Abcam), antimycin A (1 μM) (Sigma). Measured values were normalized to total protein amount quantified by a BCA quantification kit (Huaxingbio).

**Glucose uptake measurement**. Glucose consumption and lactate production were measured by LC-MS/MS. Cells were cultured in the regular medium. After 12 h, the medium was collected and added to 4 volumes of 100% methanol. Metabolites were extracted and analyzed by LC-MS/MS. Ultimate 3000 UHPLC (Dionex) coupled with HF orbitrap (Thermofisher) was used to perform LC separation. In negative mode, BEH Amide column (2.1 × 100 mm, Waters) was applied for separation with column temperature at 50 °C. The gradient is generated with flow rate at 400 μL/min as follows: 0 min, 2%B; 1 min, 2% B; 5.5 min, 10% B; 6.5 min, 80% B; 8.3 min, 2% B; 10.0 min, 2% B. Mobile phase A is prepared by dissolving 0.58 g of ammonium acetate in 50 mL of HPLC-grade water, then adding 950 mL of HPLC-grade acetonitrile. Adjust pH to 9.0 with ammonium hydroxide solution. Mobile phase B is prepared by dissolving 0.58 g of ammonium acetate in 500 mL HPLC-grade water, subsequently, adding 500 mL HPLC-grade acetonitrile and adjusting pH to 9.0 with ammonium hydroxide solution. Data with mass ranges of m/z 80–300 was acquired at negative ion mode with data-dependent MSMS acquisition. The full scan and fragment spectra were collected with resolution of 60,000 and 30,000 respectively. The detailed mass spectrometer parameters are as follows: spray voltage, 3.0 kV for negative; capillary temperature, 320 °C; heater temperature: 300 °C; sheath gas flow rate: 35; auxiliary gas flow rate:10. Metabolite identification was based on Tracefinder (Thermofisher) search with home-built database.

Also, glucose uptake measurement was performed using the Glucose Uptake-Glo™ Assay (Promega) according to the manufacturer's directions. In brief, the cell culture medium was removed, and plates were incubated in 1 mM of 2-DG in PBS for 10 min at room temperature. Following cell lysis and neutralization, samples were incubated in the 2DG6P detection reagent for 1 h and analyzed with a luminometer.

**Metabolites extraction and metabolomic analysis**. Day 3 EBs were collected in a 15 mL tube and washed gently with PBS buffer 3 times. The tubes were placed on dry ice and 2 mL of 80% (vol/vol) methanol (pre-chilled to −80 °C) were added. The tubes were centrifuged at 14,000 g for 20 min at 4–8 °C and the metabolite-containing supernatant was transferred to a new 15 mL tube on dry ice. A Speedvac (Thermo Fisher Scientific) was used to dry the pellet using no heat (room temperature). Targeted metabolomics was implemented with a TSQ Quantiva Triple Quadrupole mass spectrometer (Thermo Fisher Scientific). Mobile phase A was prepared by using 10 mM tributylamine, 15 mM acetate in HPLC-grade water, Mobile phase B was HPLC-grade 100% methanol. This analysis focused on TCA cycle, glycolysis pathway, pentose phosphate pathway, amino acids, and purine metabolism. In this experiment, we used a 25-minute gradient from 5% to 90% mobile B. Positive-negative ion switching mode was performed for data acquisition. Cycle time was set as 1 s and a total of 138 ion pairs were included. The resolution for Q1 and Q3 are both 0.7 FWHM. The source voltage was 3500 v for positive and 2500 v for negative ion mode. The source parameters are as follows: capillary temperature: 320 °C; heater temperature: 300 °C; sheath gas flow rate: 35; auxiliary gas flow rate: 10. Tracefinder 3.2 (Thermo Fisher Scientific) was applied for metabolite identification and peak integration.

For multivariate model analysis, between-group differences in metabolite abundance were assessed by Welch's two-sample t-test. Hierarchical clustering (HCL) and unsupervised principal components analysis (PCA) were implemented to assess data quality and detect sample outliers. Raw data from the metabolic analysis were normalized to the median of each sample, respectively. Missing values were assumed to be below the limit of detection and were imputed with half of the minimum of the whole dataset. The heatmap of metabolites in glycolysis, TCA cycle, and pentpentose phosphate pathway were constructed in RStudio (V1.2.1335).

**Bimolecular fluorescence complementation (BiFC)**. BiFC assays were performed as described previously[86]. Venus fluorescence protein was selected as the reporter for complementation. Mouse NEMEP including full length, ΔN, ΔC, and ΔTMD coding sequences were cloned into NheI- and XbaI- digested pcDNA3.1 vector with the amino- (YN, amino acid 1–173) termini of Venus. And the mouse GLUT1/3 coding sequences were cloned into NheI- and XbaI- digested pcDNA3.1 vector with the carboxyl- (YC, amino acids 155–239). HEK293T cells with co-transfection of YC and YN for 48 h were analyzed for fluorescence by FACS and FlowJo v7.6. For flow-based sorting of the GFP positive cells, parental cells without

plasmid transfection were used to define negative cell populations and set gates for analysis. For flow studies of BiFC, we use the same cells without plasmid transfection for defining negative cell populations and set gates for analysis.

**Quantitative reverse transcription-PCR analysis**. RNA extraction, reverse transcription, and quantitative real-time polymerase chain reaction (qPCR) were performed as described[19]. The sequence of primers used in qPCR analysis is listed in Supplementary Table 6.

**Immunoprecipitation and western blot**. Whole cell lysates were prepared from exponentially-growing cells. After centrifugation at 3000 $g$ for 5 min, cell pellets were lysed in whole cell lysis (WCL) buffer (10 mM Tris-HCl pH8.0, 150 mM NaCl, 1% Triton X-100, 5 mM EDTA) supplemented with protease inhibitors cocktail and phosphatase inhibitor cocktail (Biotool). The lysates were incubated on ice for 30 min and vortexed in 5 min intervals, at finally the lysates were cleared by centrifugation at 12,000 $g$ for 10 min at 4 °C. The soluble material was collected and incubated with Anti-Flag Affinity Gel (Biotool) or GFP-Trap Agarose (ChromoTek) for 3 h. Beads were then washed 4 times with WCL buffer, captured protein complexes were boiled in SDS sample buffer and analyzed by SDS-PAGE followed by immunoblotting. The blots were blocked with 5% non-fat powdered milk in TBST (0.1% Tween-20 in TBS) at room temperature for 1 h and incubated with primary antibodies diluted in blocking buffer overnight at 4 °C. Membranes were incubated with the appropriate secondary antibodies conjugated with horseradish peroxidase for 2 h at room temperature. Enhanced chemiluminescence system (GE healthcare, RPN2108) was used for Western blot detection. The primary antibodies used in this paper targeted the following proteins: FLAG (Sigma-Aldrich, F3165, 1:3000), GAPDH (ZSGB-BIO, TA-08, 1:5000), GFP (EASYBIO, BE2001, 1:2000), Histone H3 (Abcam, ab10799, 1:10000),SMAD2/3 (CST, 8685, 1:1000), SMAD4 (Santa Cruz, sc-7966, 1:1000), CDH1 (CST, 3195, 24E10, 1:1000), GLUT1 (Beyotime, AF1015, 1:1000), GLUT3 (Abclonal, A4137, 1:1000), HA (Millipore, 04–902, 1:2000), Phospho-SMAD2/3 (CST, 8828, 1:1000), SMAD2 (CST, 5339, 1:3000), TRIM33 (Bethyl Laboratories, A301–060A, 1:1000).

**Ribosome profiling**. $2.5 \times 10^5$ mESCs were seeded on 15-cm dish and cultured without LIF for 3 days. Cells were then treated with cycloheximide at a final concentration of 100 μg/mL for 10 min in the 37 °C 5% $CO_2$ incubator. Aspirate the media, wash the plates with ice-cold PBS containing 100 μg/mLcycloheximide (PBS/CHX). Then scrap and collect the cells with ice-cold PBS/CHX into 15 mL tubes. The cells were centrifuged at 500 $g$ for 5 min at 4 °C. Then pellet was subsequently lysed in the polysome lysis buffer (10 mM HEPES-pH 7.4, 5 mM MgCl₂, 150 mM KCl, 1% NP40, 100 μg/ml cycloheximide, 1 mM DTT, Rnase inhibitor (200 u/ml) and Protease inhibitor) and incubated on ice for 10 min. Lysates were centrifuged at 13,000 $g$ for 10 min at 4 °C, and supernatant was carefully transferred to a new 1.5 mL tube. Prepare a 15–45% linear sucrose gradient (sucrose buffer containing 10 mM HEPES-pH7.4, 5 mM MgCl2, 150 mM KCl, 100 μg/mL cycloheximide, 1 mM DTT, and sucrose.) using a Gradient Master (Gradient108, Biocomp company). Lysates were loaded on a 15–45% linear sucrose gradient and centrifuged at 40,000 $g$ for 2 h at 4 °C in a Beckman SW41 rotor and subsequently fractionated using a fraction collector (Gradient108, Biocomp company). Polysome profiles were measured using an absorbance detector connected to the fraction collector and measuring absorbance at 260 nm. RNA was extracted with Trizol. For RNA analysis from polysome fractions, qPCR were done as described[19]. The quantity of each fraction was first determined using a standard curve, and then expression was normalized to RNA contained in fraction 1.

**Quantification and statistical analysis**. The statistical tests used in this study are indicated in the respective figure legends. In general, data with two groups were analyzed by Student unpaired $t$-test to determine statistically significant effects. Data with multiple groups were analyzed by one-way, two-way ANOVA to determine statistically significant effects.

**Reporting summary**. Further information on research design is available in the Nature Research Reporting Summary linked to this article.

## Data availability

The RNA-seq data used in this study are available in the NCBI's Gene Expression Omnibus database under accession code GSE157073, GSE115169, GSE70486. The ChIP-seq data used in this study are available in the NCBI's Gene Expression Omnibus database under accession code GSE125116. The mass spectrometry data generated in this study are provided in Supplementary Data 1. Source data are provided with this paper.

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

## Acknowledgements

We thank Drs. Jianbin Wang, Jie Na, Suneng Fu, Peng Jiang, Ye-Guang Chen, Qinghua Tao, Wei Wu, and Nieng Yan for reagents, and helpful discussions. We also thanks Technology Center for Protein Sciences in Tsinghua University: Xianbin Meng, Chongchong Zhao, Xiaolin Tian, and Dr. Haiteng Deng at Protein Chemistry Facility for mass spectrometry analysis; Xueying Wang, Yupei Jiao and Dr. Xiaohui Liu at Facility Center of Metabolomics and Lipidomics for targeted metabolomic experiment analysis and LC-MS/MS analysis; Yanli Zhang and Yalan Chen at Imaging Core Facility for assistance of using Nikon A1R HD25. We would like to thank Pengcheng Jiao and Jiaojiao Ji at Center of Biomedical Analysis for flow-cytometry analysis; Jing Zhang and Dr. Zai Chang from Laboratory Animal Research Center for generating chimera embryos in Tsinghua University. We thank Ping Wu for establishing CRISPRi and CRISPRa cell lines. The work was supported by National Natural Science Foundation of China (NSFC:91540108) and Tsinghua-Peking Center for Life Sciences.

## Author contributions

H.F. and X.K. established most cell lines. H.F., X.K., and N.W. performed cell differentiation assays and qPCR. K.K. supervised N.W. for cell differentiation assays. X.K. performed the RACE assay. T.W. performed ribosome profiling. and H.F. and T.W. performed Seahorse Extracellular Flux Analysis and glucose uptake experiment. H.F. prepared RNA-seq and metabolomics samples. K.Y., J.C., and Y.Y. analyzed the RNA-seq and ChIP-seq data under the supervision of Q.X. and Z.L. Y.Y. helped to perform affinity purification assay. H.F. performed affinity purification assay, chimera embryos dissection, immunostaining,

immunoblotting, and co-IP experiment. Q.X. supervised the study. H.F. and Q.X. wrote the manuscript; all authors contributed to data interpretation and discussion.

## Competing interests

The authors declare no competing interests.
