## [Peer Review File · Nature Communications]

A Nodal enhanced micropeptide NEMEP regulates glucose uptake during mesendoderm differentiation of embryonic stem cellsREVIEWER COMMENTS

Reviewer #1 (Remarks to the Author):

Key results

In this article, Fu et al. have demonstrated that during mesendoderm differentiation, lncRNA Gm11549 is upregulated and that the micropeptide encoded by this lncRNA, NEMEP is essential for mesendoderm differentiation through activation of glycolysis by interaction with GLUT1/3. The upregulation of glycolytic activity by NEMEP is important for mesendoderm differentiation.

Validity

In principle, I have found no flaws that prohibit the manuscript's publication. However, the authors need to clarify the following points.

Significance

The finding that a micropeptide encoded by a lncRNA plays important role in mesendoderm differentiation through interaction with GLUT1/3 is significant. However, the authors need to clarify that activation of glucose transporter activity of GLUT1/3, not other mechanisms, by NEMEP is important for mesendoderm differentiation.

Data and methodology

Figure 1c

The authors argue that Gm11549 is directly regulated by TGF- β signaling since the mRNA level is not reduced by cycloheximide. It would be nice to see positive control of which the mRNA level is reduced by cycloheximide.

Extended figure 6a

Please reformat the graph.

Analytical approach

Figures 1d, Extended figures 4b

Some figures which seems to have statistically significant difference lack asterisks. Do they have no statistically significant difference? Please clarify whether there is statistically significant difference or not.

Figure 2b

It would be nice to see results from three different experiments.

Figures 5g-i, k-n, Extended Figures 6h-j

Please clarify if there are any statistically significant difference.

Figures 6e-h

Please clarify if there are any statistically significant difference. If not, please amend the following corresponding description in page 9, lines 9-11 since it can mislead the readers as if there were a significant difference in glycolytic activity.

"Compared to WT EBs, the Nemep KO EBs displayed significant reductions in glucose uptake, glycolytic function, and mitochondrial respiration."

Suggested improvements

Although the finding that NEMEP is required for mesendoderm differentiation seems to be firm,

the authors need to demonstrate that activation of glycolytic activity by NEMEP is essential for mesendoderm differentiation, not by other activity of NEMEP. The questions below should be answered by the authors.

-Although NEMEP seems to activate glycolytic activity and NEMEP interacts with GLUT1/3, the authors did not show transporter activity itself of GLUT1/3. Please clarify if NEMEP activate glucose transporter activity of GLUT1/3 or NEMEP activate glycolysis through other mechanism(s). Also, is upregulation in glycolysis cause of successful mesendoderm differentiation? Or is it just a result of successful mesendoderm differentiation?

-Is high glucose culture environment able to compensate the lack of NEMEP for mesendoderm differentiation?.

Figure 5a

-Figure 5a demonstrates that there are other membrane proteins which may interact with NEMEP. Although previous literatures demonstrate the importance of GLUT1/3 in early development as the authors described, please explain why you have chosen GLUT1/3 instead of other proteins. Does interaction with those proteins (if any) affect mesendoderm differentiation? Is there any specific reasons which exclude the possibility of involvement of other proteins?

Figure 6i

Authors showed reduction in levels of glycolytic and TCA cycle metabolites. Are metabolites in pentose phosphate pathway and latter part of TCA cycle (e.g. succinate and malate) altered by loss of NEMEP?

References

A study by Cliff et al. (PMID: 28965765) demonstrates an early switch to oxidative phosphorylation from glycolysis upon mesoderm and ectoderm differentiation compared to ectoderm in human pluripotent stem cells. Although I understand that this study was focused on mouse pluripotent stem cells, may be the authors can refer to it in discussion.

Reviewer #2 (Remarks to the Author):

Fu et al. (Xi) Nature Communications

The authors describe the identification of a lnc RNA that encodes a small transmembrane peptide that they call NEMEP (Nodal enhanced mesendoderm micropeptide) and propose its expression to be directly induced in response to TGF- β /Nodal signaling through Smad2 and/or Smad3. They carry out a set of experiments that lead them to propose a model whereby, in response to Nodal, NEMEP is expressed and is required for mesendoderm differentiation. They show that NEMEP associates with the glucose transporters Glut1 and Glut3, and propose that NEMEP promotes glucose uptake through Glut1 and Glut3, and that this occurs in response to TGF- β signaling. They also causally link the requirement of NEMEP in mesendoderm differentiation to Nodal-induced, NEMEP-dependent glucose uptake.

Overall: The authors present some interesting observations. They link these together to propose a scenario for a molecular mechanism that combines the following elements: (1) Nodal induces, through directed effects of Smad2 and/or Smad3, the expression of a lncRNA encoding NEMEP, (2) NEMEP associates with Glut1 and Glut3, (3) NEMEP controls and is required for

increased glucose uptake in response to Nodal, through its association with Glut1 or Glut3. They additionally propose that (4) NEMEP is required for mesendoderm differentiation, and that (5) this requirement for mesendoderm differentiation is explained by NEMEP's role in glucose uptake through Glut1 and Glut3.

With the exception of (2) and (4), which are not mechanistic conclusions, the other, mechanistic conclusions and causal links are insufficiently supported by the data presented, as will be further explained below. In a number of cases, developmental timelines of up to five days are an insufficient basis to conclude molecular mechanisms that are direct and more rapid, and, following synthesis of NEMEP, do not require additional transcription and translation.

Let me elaborate:

- It is aggravating to read a manuscript that mentions TGF- β , and often TGF- β /Nodal, even though the role of TGF- β or Nodal were not evaluated. I do realize that activin is often used by developmental biologists as a substitute for nodal because they act through the same receptors, but nevertheless it is activin not Nodal that is evaluated, and the conclusions relate to Nodal. Furthermore, TGF- β is very different from Nodal or activin, which act through different receptor complexes. Along the same lines, the authors often mention Smad2/3; however, Smad2/3 does not exist, and Smad2 and Smad3 are very different and act very differently. Additionally, Glut1/3, as often mentioned, needs to be Glut1 and Glut3, or Glut1 or Glut3. This cavalier and sloppy usage of the names is very disturbing. Unfortunately, it also reflects to some extent the overall tenor throughout the manuscript as it relates to making the conclusions presented.
- Related to this, the authors should not designate activin as AC in the text; nobody does this. Similarly, SB431542 is the name, not SB. I can see that Ac and SB are used in figures, but not in the text.
- The authors seem to be unaware of the substantial literature on TGF- β -induced increase in glucose uptake and metabolism, and on the ability of any agent that induces Akt activation (and that includes TGF- β family proteins) to upregulate glucose transporters at the cell surface. This literature needs to be taken into account when designing experiments and interpreting data.
- The information of how the experiments were done is often insufficient, thus making it often unclear what was done. Consequently, a reviewer cannot gauge at times whether a conclusion is justified. One particular example is apparent in my next point.

I. As to conclusion (1), i.e. Nodal induces, through direct Smad-mediated transcription, the expression of a lncRNA encoding NEMEP:

- Line 28 on page 4: SB431542 is NOT an activin antagonist. It is a kinase inhibitor that blocks activin and TGF- β -induced Smad activation (but not Akt activation).
- Fig. 1c does not tell me how the experiment is done, e.g. how long were the cells treated with activin, when was cycloheximide given with respect to activin.
- In Fig. 1b, how long was the activin treatment prior to ChIP? How does the SB431542 control look like in the ChIP experiment?
- Related to line 39: A predicted Smad2/3 binding site, as stated, does not exist, since Smad2 (at least in its most common form) does not bind DNA.
- Related to Fig. 1f and 1g: No conclusions can be made about the participation of Smad3, even though the authors state that no induction was apparent in the absence of Smad3 (middle paragraph page 5). Smad3 KD was only evaluated in a Smad2 KO setting.
- Related to Fig. 1g-k: What is the treatment regimen?
- Fig. 2b: Unclear how this is quantified.

- Fig. 2j: the nuclear fraction is not shown, even though mentioned in the legend.

In conclusion, that the lncRNA or NEMEP expression is induced as a direct target in response to Nodal or activin is not sufficiently supported. I do not know enough about the treatment. A direct induction is scored by the expression of the RNA after a short treatment with ligand and should be unaffected by cycloheximide (as maybe shown in Fig. 1b, but I do not know the treatment regimen). Additionally, while activin can induce the lncRNA and NEMEP expression, no evidence is provided that its induction in embryoid bodies is induced by Nodal. Hence, there is no evidence whatsoever that justifies the name NEMEP, with NE standing for "Nodal enhanced".

II. As to conclusion (2): Does NEMEP also interact with Glut2 and Glut4, or is this association restricted to Glut1 and Glut3? Different glucose transporters may have different roles, depending on the stage of EB differentiation. Additionally, and possibly but not surely more importantly, no functional data link NEMEP to Glut1 or Glut 3.

III. As to conclusions (3) and (5), i.e. that NEMEP controls or is required for Nodal-induced glucose uptake through Glut1 and Glut3.

- What is the effect of Glut1 or Glut3 knockout or knockdown on glucose uptake? The only evidence so far for NEMEP acting through Glut1 and/or Glut3 is coimmunoprecipitation of NEMEP with Glut1 and Glut3.

- There is no evidence that endogenous Nodal controls glucose uptake. Yes, activin can induce glucose uptake, as has been reported for TGF- β , but this does not mean that Nodal does so naturally in EBs. And then I do not even address the question whether Nodal acts through Glut1 or Glut 3, for which there is no evidence presented.

Reviewer #3 (Remarks to the Author):

In this study, Haipeng Fu et al. identify a lncRNA that is specifically expressed in response to TGF β /Nodal stimulation in mESC-derived EBs. The authors convincingly show that such lncRNA is actually able to encode for a 63aa micropeptide, which is highly conserved in human, and that they named NEMEP. Furthermore, intriguingly, the authors demonstrate that this peptide interacts with glucose transporter proteins enhancing the uptake of glucose, which, in turn, is required for mesendoderm differentiation.

Overall, this is a compelling story that depicts a new factor and its mechanism of action, which is shown to contribute to the differentiation of pluripotent cells in a surprising way.

The authors may consider addressing the following points:

Major point:

The series of experiments shown here nicely proved that expression of NEMEP results in increased glucose metabolism, which seems to be required for successful mesendoderm differentiation. It would be useful to clarify whether this boost in glucose uptake (driven by NEMEP expression) directly contributes to the mesendodermal induction, or rather it

metabolically sustains this developmental step.

In other words:

- 1) does the overexpression of NEMEP induce the expression of LDTFs in EBs-AC?
- 2) does the overexpression of NEMEP rescue the reduced expression level of mesendoderm developmental marker genes in EBs cultured at low glucose concentration? And, can high glucose concentration rescue NEMEP-KO phenotype?

Minor points:

- 1) Most of the figures are very dense and might result difficult to read in a final version of the manuscript. Could the author move some of the less informative panels to the extended data section? For example, Figure 4h, i, m and n seem to carry the same information as Figure 4g and l.
- 2) In Figure 1b the authors present the characterization of the Gm11549 transcript by 3'end RACE analysis, but it is not clear what conclusion they draw from this.
- 3) In Figure 1e and Extended F1d, the author should specify in the figure legends what is on the Y axis of these Corn Plots.
- 4) In Figure 3e and f, the promoter KO seems to have more severe phenotype than the NEMEP KO. This would indicate that, besides the small peptide, the lncRNA might exert some other function? Could the Author comment or clarify on this point?
- 5) In Figure 4, the pictures of mouse embryos used are not very good looking. Could the author include additional images, perhaps adding more mesendoderm marker genes, such as T?

Reviewer #4 (Remarks to the Author):

This manuscript provides a fairly complete set of data on how a long noncoding RNA (lncRNA) encodes a regulatory micropeptide with crucial (physiological) biological function. This is an important point because the recent discovery that lncRNAs actually encode peptides provides the scientific community with a new perspective on the regulation of gene expression. In this manuscript, the authors identified a direct target gene of transforming growth factor- β /Nodal signaling, Gm11549, and found that the lncRNA Gm11549 can be translated. They found that Gm11549 encodes a highly conserved 63 amino acid single-pass transmembrane micropeptide, NEMEP, which interacts with two glucose transporter proteins (GLUT1/3) and facilitates glucose uptake through these physical interactions. However, there are several concerns that should be addressed:

1. Leukemia Inhibitory Factor (LIF) is mentioned several times in the manuscript, please explain the role of LIF so that it can be understood by readers who are not specialized in the field.
2. In figures 1F and 1G, the author claims "both Gm11549 transcription and responsiveness of

Gm11549 to TGF- β are dependent on SMAD2/3/4 individually, as both phenotype were impaired in Smad2, Smad3 or Smad4 depletion cells". However, the authors did not verify the results by using TGF- β -treated cells, and whether the authors can supplement the experiment on the detection of Gm11549 levels in the TGF- β -treated group. In addition, is there direct evidence that TGF- β can regulate the expression of Gm11549?

3. For lncRNA Gm11549, the authors should further detect the expression and subcellular localization of Gm11549 by Northern blot and fluorescence in situ hybridization (FISH).

4. The ribosome profiling assay should be performed to detect whether lncRNA Gm11549 has translation potential.

5. In figures 2H and 2I, the author claims that Gm11549 ORF1 can encode a micropeptide in vivo. However, how can the authors exclude the effect of fluorescent background and non-specific binding of antibodies. In addition, the author also needs to detect whether the initiation codon of Gm11549 ORF1 is translationally active.

6. In Figure 3B, the authors should detect the expression level of NEMEP in each group.

7. The authors should show the interaction of endogenous NEMEP with endogenous GLUT1 and GLUT3.

8. In Figure 5E and 5F, the authors only examined glucose uptake and lactate excretion in the NEMEP overexpression group, but what was the status in the NEMEP KO group?

9. In figure 5H-N, the relevant grouping information is missing, please indicate it.

10. In Figure 6E-H, please indicate whether the relevant groups were statistically analyzed and whether there were statistical differences. Also, a similar problem occurs in other figures in the manuscript, please explain.

11. In supplementary figure 6A, the data shown in the figure seems to be incorrect. Please check it carefully.

12. More detailed materials and methods section should be included which outlines how the data was generated needs to be included. For examples how was the RNA-seq data analysed, currently unclear.

Prior to getting into our full point-by-point response, we would like to first offer our gratitude for the excellent guidance from the Editor and the reviewers about improving our study. We have now completed all of the requested experiments and made corresponding revisions to our manuscript, and we trust you'll agree that our study has been substantially improved by this revision process. As a brief executive summary of the topics addressed during our revision process, we have now:

- included additional replicates and statistics as requested (Reviewers #1, #4)
- provided functional assessment of whether NEMEP regulation of glycolysis is required for mesendoderm differentiation and clarified whether NEMEP induced glucose uptake directly contributes to mesendoderm differentiation (Reviewers #1, #2, #3)
- justified our scientific focus on GLUT1/3 (Reviewer #1)
- presented a more detailed assessment of metabolite changes (Reviewer #1)
- distinguished among TGFb, Nodal, and Activin (Reviewer #2)
- conceptually placed our study appropriately within the literature (Reviewer #2)
- clarified multiple aspects of our experimental procedures (Reviewers #2, #4)
- assessed loss of *Glut1* and *Glut3* (Reviewer #3)
- included ribosome profiling (Reviewer #4)
- confirmed the interactions of endogenous NEMEP with endogenous GLUT1 and GLUT3 (Reviewer #4)
- included missing controls and clarifications where requested.

Again, Many thanks for the ongoing guidance and support of our manuscript.

Reviewer1:

Key results

In this article, Fu et al. have demonstrated that during mesendoderm differentiation, lncRNA Gm11549 is upregulated and that the micropeptide encoded by this lncRNA, NEMEP is essential for mesendoderm differentiation through activation of glycolysis by interaction with GLUT1/3. The upregulation of glycolytic activity by NEMEP is important for mesendoderm differentiation.

Validity

In principle, I have found no flaws that prohibit the manuscript's publication. However, the authors need to clarify the following points.

Significance

The finding that a micropeptide encoded by a lncRNA plays important role in mesendoderm differentiation through interaction with GLUT1/3 is significant. However, the authors need to clarify that activation of glucose transporter activity of GLUT1/3, not other mechanisms, by NEMEP is important for mesendoderm differentiation.

Response: We would like to thank the reviewer for the careful review and for the excellent guidance about how to improve our study. Regarding this comment specifically, we thank the reviewer for pointing this out. We used Glucose Uptake-Glo™ Assays (Promega) for measuring glucose uptake (Fig. 5e, h, i, j, and Fig. 6c, e, and Fig. 7a-f). Glucose uptake occurs on a rapid time scale (of 10 minutes or less according to the protocol), and the measurement readout of the assay specifically reflects the glucose transporter activity. Thus, we interpret these results as showing that the glucose uptake changes do reflect the levels of glucose transporter activity. We have modified the manuscript to clarify this point (Page 9 Line 11).

Data and methodology

Figure 1c

The authors argue that Gm11549 is directly regulated by TGF- β signaling since the mRNA level is not reduced by cycloheximide. It would be nice to see positive control of which the mRNA level is reduced by cycloheximide.

Response: We thank the reviewer for pointing this out. We added *Lefty2* as a control; the *Lefty2* mRNA level is reduced by cycloheximide (Fig. 1c).

Figures 1d, Extended figures 4b

Some figures which seems to have statistically significant difference lack asterisks. Do they have no statistically significant difference? Please clarify whether there is statistically significant difference or not.

Response: We thank the reviewer for pointing this (and the other) omissions regarding our use of inferential statistics throughout the text. we have carefully ameliorated this problem in our revision, and consistently provide information about sample sizes and about exactly which inferential tests were used for each of the comparisons. Regarding this point specifically, we have made changes to the original Fig. 1d and Supplementary Fig. 4b (Supplementary Fig. 5d in the revised manuscript).

Figure 2b

It would be nice to see results from three different experiments.

Response: We thank the reviewer for pointing this out. We provided the data here (Supplementary Fig. 2b in the revised manuscript) is representative of three independent experiments of nuclear/cytosolic fractionation assay. We have updated our captions to accurately present this information throughout the revised manuscript.

Figures 5g-i, k-n, Extended Figures 6h-j

Please clarify if there are any statistically significant difference.

Response: We thank the reviewer for pointing this out. We have made changes accordingly to clarify statistically significant differences in the original Figures 5g-i, k-n, Supplementary Figures 6h-j accordingly (original Fig. 5h, i are now Supplementary Fig. 7b, c; original Fig. 5k is now Fig. 5i; original Fig. 5 l-n are now Supplementary Fig 8a-c; original Supplementary Figures 6h-j are now Supplementary Fig. 8e-g).

Figures 6e-h

Please clarify if there are any statistically significant difference. If not, please amend the following corresponding description in page 9, lines 9-11 since it can mislead the readers as if there were a significant difference in glycolytic activity.

Response: We thank the reviewer for pointing this out. We have made changes accordingly to clarify statistically significant differences in the figures (original Fig. 6e-h are now Fig. 6g-j in the revised manuscript).

Although NEMEP seems to activate glycolytic activity and NEMEP interacts with GLUT1/3, the authors did not show transporter activity itself of GLUT1/3. Please clarify if NEMEP activate glucose transporter activity of GLUT1/3 or NEMEP activate glycolysis through other mechanism(s).

Response: We thank the reviewer for pointing this out. In these experiments we used Glucose Uptake-Glo™ Assays (Promega) for measuring glucose uptake in both NEMEP OE, KO, cells and GLUT1/GLUT3 KD cells (Fig. 5e-h, Fig. 6c and e). Glucose uptake occurs on a rapid time scale (of 10 minutes or less according to the protocol), and the measurement readout of the assay specifically reflects the glucose transporter activity. Thus, we interpret these results as showing that GLUT1/GLUT3 do have glucose transporter activities and NEMEP does activate glucose transporter activity. We have modified the manuscript to clarify this point (Page 9 Line 11).

7. Also, is upregulation in glycolysis cause of successful mesendoderm differentiation? Or is it just a result of successful mesendoderm differentiation?

Response: After substantial consideration, if we understand the reviewer's questions correctly, then we must assume you are looking for a global consideration of the observed processes. For some context we must recall that developmental processes are hierarchically complex and display emergent properties that are dynamic in time

and space, with both master and fine-tuning-type regulatory mechanisms involved, indeed interacting at multiple layers, including transcription, post-transcription, translation, post-translation, metabolism etc. (PMID: 23715547; PMID: 29414683; PMID: 20439159; PMID: 8631492; PMID: 22868264; PMID: 10934024; PMID: 2645302; PMID: 24710195; PMID: 503184; PMID: 469836; PMID: 11180962).

Myriad studies have revealed that metabolic programs are regulated in a cell-context dependent manner by specific transcription factors or other regulators; and, reciprocally, metabolism is known to promote or inhibit fate regulators in early development. Thus, our data support the confident assertion that the upregulation in glycolysis is necessary, but is not—in and of itself—sufficient to regulate successful mesendoderm differentiation. We also now have data from a new experiment that is relevant to this comment. It is known that overexpression of GLUT1 or GLUT3 cause the upregulation of glycolysis (PMID: 20209635; PMID: 26650681). To answer this point, we established mESCs with overexpressing GLUT1 or GLUT3, and set up the mesendoderm differentiation *in vitro*. The qPCR assay showed that the mesendoderm marker genes expression significantly declined in the cells with overexpression of the GLUT1 or GLUT3 (Response Document Fig I. a, b; for reviewer only), indicating that upregulation of glycolysis is not enough to promote successful mesendoderm differentiation.

Response Document Figure I

a, Validating *Slc2a1* or *Slc2a3* expression in WT and GLUT1 or GLUT3 overexpressing mESCs by qPCR analysis.

b, GLUT1 or GLUT3 overexpressing and WT mESCs were induced for EB formation for the indicated durations. Total RNA was analyzed by qPCR using primers for the indicated genes.

8. Is high glucose culture environment able to compensate the lack of NEMEP for mesendoderm differentiation?

Response: We thank the reviewer for focusing our attention here. Glucose is essential for efficient mesendoderm differentiation (Fig. 6a, b). However, high glucose does not promote mesendoderm differentiation (Fig. 6d). We have now completed *in vitro* differentiation assays with WT and *Nemep* KO cells to which we added different glucose concentrations. The results showed that high glucose is not able to compensate for the lack of NEMEP for mesendoderm differentiation (Response Document Fig. II; for reviewer only).

Response Document Figure II

WT and *Nemep* KO EBs at day 2 were cultured in medium containing 125 mM, 25 mM glucose. Expression of the indicated genes was analyzed by qPCR at EBs day3.

Figure 5a

-Figure 5a demonstrates that there are other membrane proteins which may interact with NEMEP. Although previous literatures demonstrate the importance of GLUT1/3 in early development as the authors described, please explain why you have chosen GLUT1/3 instead of other proteins. Does interaction with those proteins (if any) affect mesendoderm differentiation? Is there any specific reasons which exclude the possibility of involvement of other proteins?

Response: We thank the reviewer for pointing this out. The list (original Fig. 5a) we present is the first-pass, potential interactome for NEMEP (using an arbitrary cutoff in the MS dataset). Our Co-IP assay data only support that GLUT1 and GLUT3 are able to form a complex with NEMEP, so we have removed information about the other putative interacting proteins from the revised manuscript to retain our focus on the function of NEMEP in glucose transportation.

Figure 6i

Authors showed reduction in levels of glycolytic and TCA cycle metabolites. Are metabolites in pentose phosphate pathway and latter part of TCA cycle (e.g. succinate and malate) altered by loss of NEMEP?

Response: We thank the reviewer for pointing this out. We have added the metabolites of the pentose phosphate pathway and the latter part of the TCA cycle to the heatmap (original Fig. 6i is now Fig. 6k in the revised manuscript).

References

A study by Cliff et al. (PMID: 28965765) demonstrates an early switch to oxidative phosphorylation from glycolysis upon mesoderm and ectoderm differentiation compared to ectoderm in human pluripotent stem cells. Although I understand that this study was focused on mouse pluripotent stem cells, may be the authors can refer to it in discussion.

Response: We thank for the reviewer for this valuable suggestion. We have added the Cliff et al. citation, as well as related studies, to the revised introduction and discussion (Page 3 Line 5, Page 12 Line 15).

Let us again thank the reviewer for the helpful guidance to improve our study.

Reviewer#2

Fu et al. (Xi) Nature Communications

The authors describe the identification of a lncRNA that encodes a small transmembrane peptide that they call NEMEP (Nodal enhanced mesendoderm micropeptide) and propose its expression to be directly induced in response to TGF- β /Nodal signaling through Smad2 and/or Smad3. They carry out a set of experiments that lead them to propose a model whereby, in response to Nodal, NEMEP is expressed and is required for mesendoderm differentiation. They show that NEMEP associates with the glucose transporters Glut1 and Glut3, and propose that NEMEP promotes glucose uptake through Glut1 and Glut3, and that this occurs in response to TGF- β signaling. They also causally link the requirement of NEMEP in mesendoderm differentiation to Nodal-induced, NEMEP-dependent glucose uptake.

Overall: The authors present some interesting observations. They link these together to propose a scenario for a molecular mechanism that combines the following elements: (1) Nodal induces, through directed effects of Smad2 and/or Smad3, the expression of a lncRNA encoding NEMEP, (2) NEMEP associates with Glut1 and Glut3, (3) NEMEP controls and is required for increased glucose uptake in response to Nodal, through its association with Glut1 or Glut3. They additionally propose that (4) NEMEP is required for mesendoderm differentiation, and that (5) this requirement for mesendoderm differentiation is explained by NEMEP's role in glucose uptake through Glut1 and Glut3.

With the exception of (2) and (4), which are not mechanistic conclusions, the other, mechanistic conclusions and causal links are insufficiently supported by the data presented, as will be further explained below. In a number of cases, developmental

timelines of up to five days are an insufficient basis to conclude molecular mechanisms that are direct and more rapid, and, following synthesis of NEMEP, do not require additional transcription and translation.

Let me elaborate:

1. It is aggravating to read a manuscript that mentions TGF-b, and often TGF-b/Nodal, even though the role of TGF-b or Nodal were not evaluated. I do realize that activin is often used by developmental biologists as a substitute for Nodal because they act through the same receptors, but nevertheless it is activin not Nodal that is evaluated, and the conclusions relate to Nodal. Furthermore, TGF-b is very different from Nodal or activin, which act through different receptor complexes.

Response: We very much appreciate the helpful guidance offered by the reviewer, and we want to be perfectly clear and that we understand the insufficiencies in our biochemical presentation in the originally submitted manuscript. We have now—guided by the excellent and highly specific suggestions—understood our errors and have comprehensively reworked the revised text to ensure that we are referring to the actual molecules we investigated. Regarding this point specifically, we apologize for the confusion. We have revised the text by replacing Nodal/TGF-b with Nodal. Demonstrating that *Gm11549* transcription is induced by Nodal, our data show that *Gm11549* transcription is dependent on Cryptic/Cripto, the well-established co-receptor of Nodal (Supplementary Fig. 1i, j).

2. Along the same lines, the authors often mention Smad2/3; however, Smad2/3 does not exist, and Smad2 and Smad3 are very different and act very differently.

Response: We again understand our errors and apologize for the confusion again. We made changes accordingly throughout the revised manuscript by writing “SMAD2 and SMAD3”.

3. Additionally, Glut1/3, as often mentioned, needs to be Glut1 and Glut3, or Glut1 or Glut3. This cavalier and sloppy usage of the names is very disturbing. Unfortunately, it also reflects to some extent the overall tenor throughout the manuscript as it relates to making the conclusions presented.

Response: Thank you for pointing this out. We made changes accordingly throughout the revised manuscript by writing “GLUT1 and GLUT3”.

4. - Related to this, the authors should not designate activin as AC in the text; nobody does this. Similarly, SB431542 is the name, not SB. I can see that Ac and SB are used in figures, but not in the text.

Response: Thank you for pointing this out. We made changes accordingly throughout the revised manuscript.

5. - The authors seem to be unaware of the substantial literature on TGF- β -induced increase in glucose uptake and metabolism, and on the ability of any agent that induces Akt activation (and that includes TGF- β family proteins) to upregulate glucose transporters at the cell surface. This literature needs to be taken into account when designing experiments and interpreting data.

Response: Thank you for this strong guidance about our omissions on this topic; we were indeed “behind the times” on developments outside of the somewhat narrow world of embryonic stem cells. In our new experimental work and in our revised manuscript, we have worked hard to integrate the information that we learned while addressing this comment.

Note that we have now completed an experiment to specifically evaluate whether depletion of NEMEP affects AKT activation in mESCs, given that AKT activation was shown to promote glucose uptake in adipocytes (PMID: 14522993; PMID: 8940145) and considering that TGF- β signaling led to Akt-TOR pathway activation in response to glucose stimulation (PMID: 19619490). Briefly, AKT activation is not impacted by depletion of NEMEP (Supplementary Fig. 9g). We have also added citations, new data, and relevant content to the revised main text.

6. - The information of how the experiments were done is often insufficient, thus making it often unclear what was done. Consequently, a reviewer cannot gauge at times whether a conclusion is justified. One particular example is apparent in my next point.

Response: We thank the reviewer for pointing this general trend out. Accordingly, we have carefully checked for this problem throughout the text and have added needed details.

7. - Line 28 on page 4: SB431542 is NOT an activin antagonist. It is a kinase inhibitor that blocks activin and TGF- β -induced Smad activation (but not Akt activation).

Response: We thank the reviewer for pointing out this error. We have modified the manuscript to correct it, as for example with “the ALK4/5/7 inhibitor SB431542” (Page 4 Line 33).

8. Fig. 1c does not tell me how the experiment is done, e.g. how long were the cells treated with activin, when was cycloheximide given with respect to activin.

Response: We thank the reviewer for pointing out the need to actually present the essential details of our experimental designs. We have added the information to the Fig. 1c caption in the revised manuscript. We have also added a lot of detail about the durations and the sequences of treatments to many additional captions.

9. In Fig. 1b, how long was the activin treatment prior to ChIP? How does the SB431542 control look like in the ChIP experiment?

Response: Activin A treatment is 2 hours for the ChIP assay. Note that we also added the data for the SB431542 treatment control in the Fig. 1b of the revised manuscript.

10. Related to line 39: A predicted Smad2/3 binding site, as stated, does not exist, since Smad2 (at least in its most common form) does not bind DNA.

Response: The ChIP-seq of Smad2/3 was done by using anti-SMAD2/3 antibody (CST, Smad2/3 (D7G7) XP® Rabbit mAb #8685). The commercial information for this product reads as follows:

“This monoclonal antibody is produced by immunizing animals with a synthetic peptide corresponding to residues surrounding His198 of human Smad2/3 protein “(<https://www.cellsignal.com/products/primary-antibodies/smad2-3-d7g7-xp-rabbit-mab/8685>).

To our understanding, this antibody recognizes SMAD3 as well as the two known isoforms of SMAD2 (*i.e.*, with or without exon3 of SMAD2) (PMID: 26905010). A recent structural study showed that a splicing isoform of SMAD2 with a deletion of exon3 (SMAD2 Δ exon3) does bind DNA (PMID: 31582430). Of particular note, this DNA-binding isoform of SMAD2 (SMAD2 Δ exon3) is highly abundant in mESCs (PMID: 15630024). Thus, the ChIP-seq signals represent SMAD2 and SMAD3 binding sites.

11. Related to Fig. 1f and 1g: No conclusions can be made about the participation of Smad3, even though the authors state that no induction was apparent in the absence of Smad3 (middle paragraph page 5). Smad3 KD was only evaluated in a Smad2 KO setting.

Response: We thank the reviewer for focusing our attention here. Please see Fig. 1f and 1g in the revised manuscript: we now have data showing that knockdown of Smad3 significantly impairs induction of *Gm11549*.

12. - Related to Fig. 1g-k: What is the treatment regimen?

Response: We have added the essential information to the revised manuscript (Fig. 1g-k legend).

13. - Fig. 2b: Unclear how this is quantified.

Response: Please kindly note that the original Fig. 2b is now Supplementary Fig. 2b. The detailed quantification method is in the **Methods** part (Page 21 Line 41).

14. - Fig. 2j: the nuclear fraction is not shown, even though mentioned in the legend.

Response: We apologized that we mislabeled the figure in the originally submitted manuscript. We have corrected this error in the revised manuscript (Fig. 2j legend) (Page14 Line 30).

15. In conclusion, that the lncRNA or NEMEP expression is induced as a direct target in response to Nodal or activin is not sufficiently supported. I do not know enough about the treatment. A direct induction is scored by the expression of the RNA after a short treatment with ligand and should be unaffected by cycloheximide (as maybe shown in Fig. 1b, but I do not know the treatment regimen).

Response: We apologize for the omission of this essential information for the treatment regimen. We have now added this information to the captions for Fig. 1b and Fig.1c in the revised manuscript. Moreover, we have now added a new paragraph at the very start of the results section of the revised manuscript that provides explicit context about the experimental necessity of using Activin A, “Activin A was used as a substitute for Nodal in our study because it is easier-to-obtain and because these two protein ligands act through the same receptors; one notable distinction is that Nodal requires the co-receptors *Cryptic* and *Cripto*; Activin A does not .”

Addressing the ligand aspect of this comment specifically, we measured (qPCR) the *Gm11549* expression level in EBs (with shRNA-mediated knockdown of the Nodal co-receptors *Cripto* and *Cryptic*). Compared to WT EBs, the *Cripto/Cryptic*-KD EBs had significantly reduced *Gm11549* expression (Supplementary Fig. 1i, j), supporting that Nodal does induce transcription of *Gm11549*.

16. Additionally, while activin can induce the lncRNA and NEMEP expression, no evidence is provided that its induction in embryoid bodies is induced by Nodal. Hence, there is no evidence whatsoever that justifies the name NEMEP, with NE standing for “Nodal enhanced”.

Response: We checked the expression of *Gm11549* in shRNA-mediated depletion of Nodal co-receptor *Cripto* and *Cryptic* EBs. The *Gm11549* expression is strongly blunted in *Cripto/Cryptic*-depleted EBs (Supplementary Fig. 1i, j). Thus, we would still argue that our data supporting that *NEMEP* transcription is induced as a direct consequence of Nodal-mediated activation does justify the use of the “NEMEP” name. Nevertheless, we would consider use an alternative term if one more suitable was offered.

17. Does NEMEP also interact with Glut2 and Glut4, or is this association restricted to Glut1 and Glut3? Different glucose transporters may have different roles, depending on the stage of EB differentiation. Additionally, and possibly but not surely more importantly, no functional data link NEMEP to Glut1 or Glut 3.

Response: We thank the reviewer for focusing our attention here. We now have new data of bimolecular fluorescence complementation (BiFC) analysis validated the interaction between NEMEP with four class I facilitative glucose transporters including GLUT1 to GLUT4 through fluorescence-activated cell sorting (FACS) (Supplementary Fig. 6a).

In the revised Fig. 5, we show that NEMEP facilitates glucose uptake in a manner dependent on its interaction with at least one GLUT protein (we have data for both GLUT1 and GLUT3 dependence): That is,

1a) We showed that expressing NEMEP in mESCs enhances glucose uptake (Fig. 5e-h);

1b) glucose uptake is not enhanced upon the expression of NEMEP mutant variants that are incapable of interacting with GLUT1 and with GLUT3 (Fig. 5h);

2a) The extent of glucose uptake enhancement upon GLUT1 overexpression is further significantly elevated by co expression of NEMEP. Note that no such elevation occurs with co-expression of the NEMEP mutant variants that are incapable of interacting with GLUT1. (Fig. 5i, j);

2b) The same trends were evident in experiments with GLUT3 (Fig. 5i, j).

18. What is the effect of Glut1 or Glut3 knockout or knockdown on glucose uptake? The only evidence so far for NEMEP acting through Glut1 and/or Glut3 is coimmunoprecipitation of NEMEP with Glut1 and Glut3.

Response: We thank the reviewer for pointing this out. We now have new data about the effects of Glut1 and Glut3 knockdown on glucose uptake (Fig. 6c). Briefly, *Glut1* and *Glut3* knockdown cells exhibit severe glucose uptake defects (Fig. 6c), as well as impaired mesendoderm differentiation (Fig. 6b).

19. There is no evidence that endogenous Nodal controls glucose uptake. Yes, activin can induce glucose uptake, as has been reported for TGF- β , but this does not mean that Nodal does so naturally in EBs. And then I do not even address the question whether Nodal acts through Glut1 or Glut 3, for which there is no evidence presented.

Response: We thank the reviewer for pointing this out. We have now checked the glucose uptake in both *Cripto* and *Cryptic* double knock-down EBs: for both of these mutant cell types, the glucose uptake level is significantly reduced compared to the WT (Fig. 7b), additional lines of evidence supporting that glucose uptake is regulated by Nodal.

Regarding the second part of this comment, kindly see our explanation about the apparent miscommunication in our response to the previous comment. We trust that

the reviewer does agree that we do have evidence supporting that Nodal's transcriptional induction of *Gm11549* does regulate GLUT1/GLUT3-mediated glucose uptake.

Let us again thank the reviewer for the helpful guidance to improve our study.

Reviewer #3 (Remarks to the Author):

In this study, Haipeng Fu et al. identify a lncRNA that is specifically expressed in response to TGF β /Nodal stimulation in mESC-derived EBs. The authors convincingly show that such lncRNA is actually able to encode for a 63aa micropeptide, which is highly conserved in human, and that they named NEMEP. Furthermore, intriguingly, the authors demonstrate that this peptide interacts with glucose transporter proteins enhancing the uptake of glucose, which, in turn, is required for mesendoderm differentiation.

Overall, this is a compelling story that depicts a new factor and its mechanism of action, which is shown to contribute to the differentiation of pluripotent cells in a surprising way.

The authors may consider addressing the following points:

Major point:

1. does the overexpression of NEMEP induce the expression of LDTFs in EBs-AC?

Response: First, we would like to thank the reviewer for these supportive comments and the excellent guidance about improving our study. Regarding this comment specifically, overexpression of NEMEP inhibits transcription of LDTFs, and overexpression of GLUT1 or GLUT3 also inhibits LDTFs transcription during mesendoderm differentiation (Response Document Figure III. a-c).

Response Document Figure III

a, WT and CRISPR-dCas9-VP64 (CRISPRa) mediated NEMEP overexpressing mESCs were induced for EB formation for the indicated durations. Total RNA was analyzed by qPCR using primers for the indicated genes.

b, Validating *Slc2a1* or *Slc2a3* expression in WT and GLUT1 or GLUT3 overexpressing mESCs by qPCR analysis.

c, WT and GLUT1 or GLUT3 overexpressing mESCs were induced for EB formation for the indicated durations. Total RNA was analyzed by qPCR using primers for the indicated genes.

2. does the overexpression of NEMEP rescue the reduced expression level of mesendoderm developmental marker genes in EBs cultured at low glucose concentration? And, can high glucose concentration rescue NEMEP-KO phenotype?

Response: Our data show that overexpression of NEMEP does not rescue the reduced expression level of mesendoderm developmental marker genes in EBs cultured at low glucose concentration (Response Document Figure IV, only for reviewer); however, overexpression of NEMEP does rescue the defects of glucose uptake in NEMEP depleted cells (Fig. 7f).

Moreover, we observed mesendoderm differentiation defects i) upon overexpression of NEMEP (Response Document Figure III) and ii) upon growth of EBs in a high glucose culture condition (125mM) (Fig. 6d).

Finally, we noted that the high glucose concentration did not rescue the NEMEP-KO phenotype (*i.e.*, mesendoderm differentiation defects). These results suggest that NEMEP directly impacts glucose uptake, which can subsequently affect mesendoderm differentiation (with this latter activity being indirect).

Response Document Figure IV

WT, *Nemep* KO, and NEMEP-FLAG overexpressing WT and *Nemep*KO EBs at day 2 were cultured in medium containing 25 mM, 5 mM glucose. Expression of the indicated genes was analyzed by qPCR at EBs day3.

Minor points:

1) Most of the figures are very dense and might result difficult to read in a final version of the manuscript. Could the author move some of the less informative panels to the Supplementary section? For example, Figure 4h, i, m and n seem to carry the same information as Figure 4g and l.

Response: We thank for the reviewer for this valuable suggestion. Accordingly, we have moved multiple panels into the Supplementary section in our revised manuscript (for example, old Fig. 5h, i are now Supplementary Fig. 7 b, c. And example 2, old Figures 5 l-n are now Supplementary Fig. 8a-c).

2) In Figure 1b the authors present the characterization of the *Gm11549* transcript by 3'end RACE analysis, but it is not clear what conclusion they drawn from this.

Response: We thank the reviewer for pointing out this lack of clarity. For context, we used a RACE assay to define the full-length transcript of *Gm11549* in mESCs. Note that the *Gm11549* transcript we found in EBs differs from published sequence (GI: 100503068, NM_001384269.1). We have added this information to the revised manuscript (Page 5 Line 8). That was the sole rationale and conclusion from this experiment.

3) In Figure 1e and Extended Figure 1d, the author should specify in the figure legends what is on the Y axis of these Corn Plots.

Response: We thank the reviewer for pointing out this omission. We have corrected this and now label the Y axis of the Corn Plots in the revised manuscript.

4) In Figure 3e and f, the promoter KO seems to have more severe phenotype than the NEMEP KO. This would indicate that, besides the small peptide, the lncRNA might exert some other function? Could the Author comment or clarify on this point?

Response: Thanks for inviting us to consider this in greater depth. Our initial focus here was on the following finding: the promoter KO and NEMEP KO mutations each result in defects of mesendoderm differentiation monitored by qPCR or RNA-seq and IF of mesendoderm markers. However, kindly note that we also have data showing no defects in mesendoderm differentiation result from depletion of exon2 or exon3. These results exclude the idea that the full-length *Gm11549* lncRNA may confer some mesendoderm-differentiation-related regulatory impact. Nevertheless, it could be the case that the difference in the severity in the mesendoderm differentiation phenotypes noted by the reviewer could be mediated by the *Gm11549* coding region (for the 63 aa NEMEP product) and/or the 5' UTR (Supplementary Fig. 4a and 5a, b).

5) In Figure 4, the pictures of mouse embryos used are not very good looking. Could the author include additional images, perhaps adding more mesendoderm marker genes, such as T?

Response: We have now included better images from a repeated analysis of FOXA2 immunostaining and new images of immunostaining against the mesendoderm marker T (revised Fig. 4c, d).

We would again like to thank the reviewer for the encouragement and the insightful guidance about how to make our study better.

Reviewer #4 (Remarks to the Author):

This manuscript provides a fairly complete set of data on how a long noncoding RNA (LncRNA) encodes a regulatory micropeptide with crucial (physiological) biological function. This is an important point because the recent discovery that lncRNAs actually encode peptides provides the scientific community with a new perspective on the regulation of gene expression. In this manuscript, the authors identified a direct target gene of transforming growth factor- β /Nodal signaling, Gm11549, and found that the lncRNA Gm11549 can be translated. They found that Gm11549 encodes a highly conserved 63 amino acid single-pass transmembrane micropeptide, NEMEP, which interacts with two glucose transporter proteins (GLUT1/3) and facilitates glucose uptake through these physical interactions. However, there are several concerns that should be addressed:

1. Leukemia Inhibitory Factor (LIF) is mentioned several times in the manuscript, please explain the role of LIF so that it can be understood by readers who are not specialized in the field.

Response: We thank the reviewer for pointing out this omission about the protocols we used. For context, a commonly used growth condition for mESCs supplements media with Leukemia Inhibitory Factor (LIF), which is known to promote ESC self-renewal. Withdrawing LIF impairs the stemness of mESCs. We have added this context information to the revised manuscript (Page 4 Line 21).

2. In figures 1F and 1G, the author claims "both Gm11549 transcription and responsiveness of Gm11549 to TGF- β are dependent on SMAD2/3/4 individually, as both phenotype were impaired in Smad2, Smad3 or Smad4 depletion cells". However, the authors did not verify the results by using TGF- β -treated cells, and whether the authors can supplement the experiment on the detection of Gm11549 levels in the TGF- β -treated group. In addition, is there direct evidence that TGF- β can regulate the expression of Gm11549?

Response: We apologize for this confusion. Previous studies have demonstrated that mESCs express ALK4, ALK7, ActR-II, and ActR-IIB and produce autocrine Nodal, but produce little TGF β RII (PMID: 15703277). Thus, in the mouse early embryo development field Nodal is widely viewed as the TGF- β family ligand which triggers the phosphorylation-based activation of SMAD2 and SMAD3.

In the originally submitted manuscript, we mistakenly wrote “responsivity of Gm11549 to TGF- β ”; to clarify, what we want to convey is “responsivity of Gm11549 to Nodal signaling”. In our experiments, we used Activin A as a substitute for Nodal because it is known to act on the same receptors; it must be noted that Nodal (but not Activin A) requires participation of the co-receptors Cryptic and Cripto. We have made the germane corrections throughout the revised manuscript. That is, our revised text does not make any claims about TGF- β , and now presents our argument in terms of Nodal signaling. (Supplementary Fig. 1i, j).

In addition, is there direct evidence that TGF- β can regulate the expression of Gm11549?

Response: We apologize for the confusion about Nodal and TGF- β nomenclature that we used in the originally submitted manuscript. Please see our Response to point 2, and recall that we have changed narrowed the scope of the nomenclature in our revised manuscript. To be clear: we show evidence that Nodal and Activin A can regulate *Gm11549* expression; we have no evidence that TGF- β regulates *Gm11549* expression.

3. For lncRNA Gm11549, the authors should further detect the expression and subcellular localization of Gm11549 by Northern blot and fluorescence in situ hybridization (FISH).

Response: We thank the reviewer for pointing this out. We have now completed RNA FISH analysis: our data demonstrate the cytosolic localization of *Gm11549* (Supplementary Fig. 2a, b).

4. The ribosome profiling assay should be performed to detect whether lncRNA Gm11549 has translation potential.

Response: We thank the reviewer for pointing this out. The polysome profile analysis of EBs at day3 showed that *Gm11549* is strongly associated with polysome, as well as coding gene *Gsc*, whereas non-coding RNA *H19* is associated with monosome (Fig. 2e). This provided another evidence that *Gm11549* has translation potential.

5. In figures 2H and 2I, the author claims that Gm11549 ORF1 can encode a micropeptide in vivo. However, how can the authors exclude the effect of fluorescent

background and non-specific binding of antibodies. In addition, the author also needs to detect whether the initiation codon of Gm11549 ORF1 is translationally active.

Response: We thank the reviewer for pointing this out. We used secondary antibody alone as a negative control to exclude the effect of fluorescent background and non-specific binding of antibodies (Supplementary Fig. 3b). And we have mutated the initiation codon of ORF1 in *Gm11549* full length RNA to check the translationally activity, ORF1 can only be expressed when a complete initiation codon is present (Supplementary Fig. 3a, b).

6. In Figure 3B, the authors should detect the expression level of NEMEP in each group.

Response: We thank the reviewer for pointing this out. We added the data of the expression level of NEMEP in each group in Supplementary Fig. 4d.

7. The authors should show the interaction of endogenous NEMEP with endogenous GLUT1 and GLUT3.

Response: We thank the reviewer for pointing this out. We indeed performed endogenous Co-IP to show the interaction of endogenous NEMEP with endogenous GLUT1 and GLUT3 (Fig. 5c). Briefly, we established a 3xFLAG knock in at the C-terminal of NEMEP mESC line. Co-IP assay using anti-FLAG antibody showed that the endogenous NEMEP-3xFLAG interacts with endogenous GLUT1 and GLUT3.

8. In Figure 5E and 5F, the authors only examined glucose uptake and lactate excretion in the NEMEP overexpression group, but what was the status in the NEMEP KO group?

Response: We thank the reviewer for pointing this out. NEMEP KO impairs glucose uptake (Fig. 6e). ECAR (extracellular acidification rate) of seahorse in Fig. 6f-j indicate the lactate excretion of NEMEP KO groups with presence of glucose in the medium are significantly lower compared with WT group.

9. In figure 5H-N, the relevant grouping information is missing, please indicate it.

Response: We thank the reviewer for pointing this out. We added this information in the revised manuscript. The original Fig. 5h, i are now Supplementary Fig. 7b, c; original Fig. 5k is now Fig. 5i; original Fig. 5 l-n are now Supplementary Fig 8a-c; in the revised manuscript.

10. In Figure 6E-H, please indicate whether the relevant groups were statistically analyzed and whether there were statistical differences. Also, a similar problem occurs in other figures in the manuscript, please explain.

Response: We thank the reviewer for pointing this out. We have made changes accordingly to clarify statistically significant difference for the figures in the revised manuscript.

11. In supplementary figure 6A, the data shown in the figure seems to be incorrect. Please check it carefully.

Response: We thank the reviewer for pointing this out. We have made the correction in the revised manuscript. The original **Supplementary figure 6A** is now **Supplementary Fig. 7a** in the revised manuscript.

12. More detailed materials and methods section should be included which outlines how the data was generated needs to be included. For examples how was the RNA-seq data analyzed, currently unclear.

Response: We thank the reviewer for pointing this out. We have added the detail RNA seq analysis and other detailed information in the revised manuscript.

We would again like to thank the reviewer for the encouragement and the insightful guidance about how to make our study better.

REVIEWER COMMENTS

Reviewer #1 (Remarks to the Author):

Overall, the authors have answered the questions I have pointed out sincerely. I would like to know how the authors have interpreted the finding that high glucose condition could not compensate the lack of NEMEP for mesendoderm differentiation. If NEMEP acts solely as an activator of GLUT1 and GLUT3, high glucose condition should be sufficient for successful mesendoderm differentiation.

Reviewer #2 (Remarks to the Author):

Fu et al. (Xi) Nature Communications REVISED

The authors describe the identification of a lnc RNA that encodes a small transmembrane peptide that they call NEMEP (Nodal enhanced mesendoderm micropeptide) and propose its expression to be directly induced in response to TGF- β /Nodal signaling through Smad2 and/or Smad3. They carry out a set of experiments that lead them to propose a model whereby, in response to Nodal, NEMEP is expressed and is required for mesendoderm differentiation. They show that NEMEP associates with the glucose transporters Glut1 and Glut3, and propose that NEMEP promotes glucose uptake through Glut1 and Glut3, and that this occurs in response to TGF- β signaling. They also causally link the requirement of NEMEP in mesendoderm differentiation to Nodal-induced, NEMEP-dependent glucose uptake.

The revised manuscript has been improved when compared to the previous version. I compliment the authors for the substantial amount of work that went into this research and for improving the manuscript in response to the critiques.

The proposed scenario for a molecular mechanism combines the following elements: (1) Nodal induces, through directed effects of Smad2 and/or Smad3, the expression of a lncRNA encoding NEMEP, (2) NEMEP associates with Glut1 and Glut3, (3) NEMEP controls and is required for increased glucose uptake in response to Nodal, through its association with Glut1 or Glut3. They additionally propose that (4) NEMEP is required for mesendoderm differentiation, and that (5) this requirement for mesendoderm differentiation is explained by NEMEP's role in glucose uptake through Glut1 and Glut3. Not all conclusions are sufficiently supported. Some issues need to be (better) addressed and additional experiments are required to allow for the conclusions. In this context, that NEMEP mediates nodal/activin-induced increase in glucose uptake is contradicted by the data. Let me elaborate in order of appearance of the text and data in the manuscript.

- line 144-147 (and more broadly the entire paragraph): The statement that ChIP-Seq showed SMAD2 and SMAD3 binding sites in the Gm11549 locus is not correct since an antibody against both Smad2 and Smad3 was used, and the main Smad2 variant is unable to bind DNA directly. So, you cannot state that the data show Smad2 and Smad3 binding. It may be only Smad2 or Smad3. (Note that SMAD2 and SMAD3 refer to the human proteins, while Smad2 and Smad3 refer to the mouse proteins, unless this was changed with the revision of the nomenclature)

- That Gm11549 is a direct Nodal target gene and not a target gene for another activin-like ligand is only supported by Suppl Fig. 1j, i.e. depletion of Cripto and Criptic. Is there anything I overlook? Is there additional evidence? If not, Suppl Fig. 1j should be shown as part of Fig. 1.

- lines 219-225: as mentioned by the authors the human ortholog of the Gm11549 gene is the TMEM155 gene, which is predicted to encode a 130 aa protein that, like NEMEP, is a single transmembrane protein. The authors should be more clear as to how NEMEP relates (or not, presumably) to the proposed TMEM155 protein, especially since that protein has an extensive Wikipedia page (and this is the first place to go to for many students, unfortunately).

- paragraph starting line 328 and Fig. 5h: The authors should note that the deletion of the N-terminal sequence does not affect the activity of NEMEP in this assay. How do we interpret this?

- paragraph starting line 336, and last paragraph of this section: The authors' conclusion that NEMEP's interactions with GLUT1 and GLUT3 synergistically boost glucose uptake and glycolysis is not supported by the data (Fig. 5i, j and Suppl. Fig. 8a-h). The authors only show that NEMEP enhances the activity of overexpressed GLUT1 and GLUT3, but do not show the activity in these assays of NEMEP by itself (and this is essential!). Suppose that NEMEP has a similar activity as NEMEP + GLUT1 (or GLUT3); this would immediately bring down that conclusion. Additionally, synergy requires more than an additive effect. So, the bottom line is that this conclusion is not supported by the insufficient data.

- lines 370-372: This sentence formulates the conclusions based on the data in Fig. 6. I do not agree with this conclusion; the data are overinterpreted. The data do show the roles of glucose, GLUT1 and/or GLUT3, and NEMEP in mesendoderm differentiation. However, they do not allow the conclusion that "the interaction of NEMEP with GLUT1/GLUT3 may support mesendoderm differentiation by facilitating glucose uptake", as concluded. Yet, the data are consistent with such hypothesis.

- section lines 378-393: This major section is problematic.

(1) Panel a shows the relative glucose uptake in response to activin, but the effects of the knockouts and knockdowns (panels b, c and d) on the activin-induced responses are not evaluated. So, data without and with activin need to be shown as in panel a to allow any conclusion on the roles of Cripto/Criptic, Smad2/Smad3 or TRIM33 in the activin response.

(2) The authors conclude that Nodal-induced glucose uptake does not require regulation of the transcriptome of glucose metabolism (including Slc2a1 and Slc2a3), based on the data in Suppl Fig 9i. However, I cannot see how these data allow for that conclusion. Furthermore, while SB431542 may not have a substantial effect, how do you then explain the effect of Smad2/Smad3 KD/KO in Fig. 7c.

(3) Considering the current knowledge that Akt is activated by TGF- β -related proteins and that Akt can promote glucose uptake, the authors should evaluate the effects of Akt inhibition in an assay like Fig 7a.

- section lines 394-410 (last section of Results): The data in Fig. 7f make the authors conclude that NEMEP is required for Nodal-induced upregulation of glucose uptake, a central conclusion of this manuscript. However, the data clearly argue against this conclusion! In the absence of

NEMEP, the fold induction in response to activin remains the same as in control cells. The overall levels are lower in both control and activin-treated EBs. So, I would argue that NEMEP may enhance the basal and induced levels, and thus facilitates glucose import, but does not account for the activin- or nodal-induced increase.

Taken together, I do agree with the roles of glucose and NEMEP in mesendoderm differentiation, that NEMEP expression is induced in response to Nodal, and that it interacts with GLUT1 and GLUT3. I do, however, not agree that NEMEP accounts for the activin/nodal-induced enhancement of glucose import, since the fold induction is not affected by the absence of NEMEP. Rather, it seems that NEMEP may have a role as facilitator of glucose import, likely by cooperating with other facilitators, some of which may be induced by Smad2/3 activation. Whether this facilitating role occurs through association with GLUT1 and/or GLUT3 is not known but may be assumed.

Minor:

- Abstract: The TGF- β superfamily is by its very definition a family, not a superfamily. Many scientists propagate the name superfamily, even though they should not.

- line 150, last word: "the" should be "a".

Reviewer #3 (Remarks to the Author):

With this revision, the authors have addressed the requested changes to the manuscript which has significantly improved its impact. IN my opinion, no additional steps are required at this stage. Therefore, I can now recommend this article for publication.

Reviewer #4 (Remarks to the Author):

The authors have adequately responded to all my previous concerns.

Prior to getting into our full point-by-point response, we sincerely thank Editor and reviewers for their detailed feedback. Their constructive comments helped to improve the quality of the study. We have now completed all the requested experiments and made corresponding revisions to our manuscript, and we trust you'll agree that our study has been substantially improved by this revision process.

Reviewer #1 (Remarks to the Author):

Overall, the authors have answered the questions I have pointed out sincerely. I would like to know how the authors have interpreted the finding that high glucose condition could not compensate the lack of NEMEP for mesendoderm differentiation. If NEMEP acts solely as an activator of GLUT1 and GLUT3, high glucose condition should be sufficient for successful mesendoderm differentiation.

Response: We thank the reviewer for pointing this out. Our study demonstrated that the capacity of GLUT1 and GLUT3 of glucose transportation is facilitated by NEMEP (revised Fig. 5i, j and Supplementary Fig. 8a-h). Hence, NEMEP is a facilitator of glucose transportation for GLUT1 and GLUT3. We do not exclude the possibility that NEMEP might have other functions rather than solely as a facilitator of GLUT1 and GLUT3, which could also contribute to promote mesendoderm differentiation. This worth the further investigation in the near future.

We have now performed *in vitro* differentiation assays with different glucose concentrations by using WT and GLUT1/ GLUT3 knock-down cells. The results showed that high glucose is not able to compensate for the absence of GLUT1 and GLUT3 for mesendoderm differentiation (Response Document Fig. I; for reviewer only). This result suggests that when the active transporter is impaired (i.e., knock-down GLUT1 and GLUT3, NEMEP KO), a large amount of glucose in the medium is not sufficient to transport adequate glucose into the cells.

Response Document Fig. I

WT or GLUT1/GLUT3 DKD EBs at day 2 were cultured in medium containing 125 mM, 25 mM glucose. Expression of the indicated genes was analyzed by qPCR at EBs day3.

Reviewer #2 (Remarks to the Author):

We thank the reviewer for the positive and detailed assessment of our work and for their thoughtful and constructive comments that have helped us improve our manuscript.

Fu et al. (Xi) Nature Communications REVISED

The authors describe the identification of a lncRNA that encodes a small transmembrane peptide that they call NEMEP (Nodal enhanced mesendoderm micropeptide) and propose its expression to be directly induced in response to TGF- β /Nodal signaling through Smad2 and/or Smad3. They carry out a set of experiments that lead them to propose a model whereby, in response to Nodal, NEMEP is expressed and is required for mesendoderm differentiation. They show that NEMEP associates with the glucose transporters Glut1 and Glut3, and propose that NEMEP promotes glucose uptake through Glut1 and Glut3, and that this occurs in response to TGF- β signaling. They also causally link the requirement of NEMEP in mesendoderm differentiation to Nodal-induced, NEMEP-dependent glucose uptake.

The revised manuscript has been improved when compared to the previous version. I compliment the authors for the substantial amount of work that went into this research and for improving the manuscript in response to the critiques.

The proposed scenario for a molecular mechanism combines the following elements: (1) Nodal induces, through directed effects of Smad2 and/or Smad3, the expression of a lncRNA encoding NEMEP, (2) NEMEP associates with Glut1 and Glut3, (3) NEMEP controls and is required for increased glucose uptake in response to Nodal, through its association with Glut1 or Glut3. They additionally propose that (4) NEMEP is required for mesendoderm differentiation, and that (5) this requirement for mesendoderm differentiation is explained by NEMEP's role in glucose uptake through Glut1 and Glut3. Not all conclusions are sufficiently supported. Some issues need to be (better) addressed and additional experiments are required to allow for the conclusions. In this context, that NEMEP mediates nodal/activin-induced increase in glucose uptake is contradicted by the data. Let me elaborate in order of appearance of the text and data in the manuscript.

1.- line 144-147 (and more broadly the entire paragraph): The statement that ChIP-Seq showed SMAD2 and SMAD3 binding sites in the Gm11549 locus is not correct since an antibody against both Smad2 and Smad3 was used, and the main Smad2 variant is unable to bind DNA directly. So, you cannot state that the data show Smad2 and Smad3 binding. It may be only Smad2 or Smad3. (Note that SMAD2 and SMAD3 refer to the human proteins, while Smad2 and Smad3 refer to the mouse proteins, unless this was changed with the revision of the nomenclature)

Response: We would like to thank the reviewer for the careful review. We now have changed the “SMAD2 and SMAD3” to “SMAD2 or SMAD3”.

We used SMAD2 and SMAD3 for the mouse proteins according to International Protein Nomenclature Guidelines provided in the NCBI web site: “For vertebrates, use an all uppercase gene symbol in a protein name.”. Please see the link below :

https://www.ncbi.nlm.nih.gov/genome/doc/internatprot_nomenguide/#2-formats-for-protein-names

2.- That Gm11549 is a direct Nodal target gene and not a target gene for another activin-like ligand is only supported by Suppl Fig. 1j, i.e. depletion of Cripto and Criptic. Is there anything I overlook? Is there additional evidence? If not, Suppl Fig. 1j should be shown as part of Fig. 1.

Response: We thank the reviewer for pointing this out. We have now re-organized Fig.1 and Supplementary Fig. 1. And we put the original Suppl Fig. 1j in the revised manuscript Fig.1i.

3.- lines 219-225: as mentioned by the authors the human ortholog of the Gm11549 gene is the TMEM155 gene, which is predicted to encode a 130 aa protein that, like NEMEP, is a single transmembrane protein. The authors should be more clear as to how NEMEP relates (or not, presumably) to the proposed TMEM155 protein, especially since that protein has an extensive Wikipedia page (and this is the first place to go to for many students, unfortunately).

Response: We thank the reviewer for pointing this out. We showed that *Gm11549* and *TMEM155* are highly conserved in gene locus, DNA sequence (lines 173-178 in the revised manuscript, Fig. 2a), and the amino acids sequence of the first open reading frame (ORF1) (lines 207-213 in the revised manuscript, Fig. 2d). Moreover, we have validated the expression of both mORF1 and hORF1 in mammalian cells (Fig. 2c, Supplementary Fig. 3d). Altogether, our results demonstrated that *Gm11549* and human ortholog *TMEM155* are conserved and be translated into 63aa proteins.

Noteworthy, this is the first time to use experiment to validate the translation potential of TMEM155 (Supplementary Fig. 3d).

4.- paragraph starting line 328 and Fig. 5h: The authors should note that the deletion of the N-terminal sequence does not affect the activity of NEMEP in this assay. How do we interpret this?

Response: We thank the reviewer for pointing this out. The N-terminal deletion mutant was generated by deleting the five amino acids behind the start codon of NEMEP (Fig. 5d). And this mutant protein binds with GLUT1 and GLUT3 as well as

the WT NEMEP (Fig. 5h), and it enhances the glucose uptake similar to WT NEMEP. These results suggested that removing the five amino acids at the N-terminus is not essential for NEMEP's function in regulating glucose uptake.

5.- paragraph starting line 336, and last paragraph of this section: The authors' conclusion that NEMEP's interactions with GLUT1 and GLUT3 synergistically boost glucose uptake and glycolysis is not supported by the data (Fig. 5i, j and Suppl. Fig. 8a-h). The authors only show that NEMEP enhances the activity of overexpressed GLUT1 and GLUT3, but do not show the activity in these assays of NEMEP by itself (and this is essential!). Suppose that NEMEP has a similar activity as NEMEP + GLUT1 (or GLUT3); this would immediately bring down that conclusion. Additionally, synergy requires more than an additive effect. So, the bottom line is that this conclusion is not supported by the insufficient data.

Response: We thank the reviewer for the helpful guidance to improve our study. Please see our revised Fig. 5i-j, Supplementary Fig. 8a-h.

6.- lines 370-372: This sentence formulates the conclusions based on the data in Fig. 6. I do not agree with this conclusion; the data are overinterpreted. The data do show the roles of glucose, GLUT1 and/or GLUT3, and NEMEP in mesendoderm differentiation. However, they do not allow the conclusion that "the interaction of NEMEP with GLUT1/GLUT3 may support mesendoderm differentiation by facilitating glucose uptake", as concluded. Yet, the data are consistent with such hypothesis.

Response: We thank the reviewer for focusing our attention here. We made changes accordingly in the revised manuscript by using "These results indicate that the NEMEP may support mesendoderm differentiation by facilitating glucose uptake. "

- section lines 378-393: This major section is problematic.

7. Panel a shows the relative glucose uptake in response to activin, but the effects of the knockouts and knockdowns (panels b, c and d) on the activin-induced responses are not evaluated. So, data without and with activin need to be shown as in panel a to allow any conclusion on the roles of *Cripto/Cryptic*, *Smad2/Smad3* or *TRIM33* in the activin response.

Response: We thank the reviewer for pointing this out. We have now added the data to show the relative glucose uptake level in WT, *Cripto* KD/*Cryptic* KD, *Smad2* KO/*Smad3* KD, and *Trim33* KO cells with or without Activin A treatment in the revised manuscript Fig. 7b-d.

8. The authors conclude that Nodal-induced glucose uptake does not require regulation of the transcriptome of glucose metabolism (including *Slc2a1* and *Slc2a3*), based on the data in Suppl Fig 9i. However, I cannot see how these data allow for that conclusion. Furthermore, while SB431542 may not have a substantial effect, how do you then explain the effect of *Smad2/Smad3* KD/KO in Fig. 7c.

Response: We thank the reviewer for pointing this out. We apologize for the confusion. We now revised the manuscript: “However, given that the expression of well-known glucose metabolism genes (including *Slc2a1* and *Slc2a3*) was not altered by activation (Activin A treatment) or inhibition (SB431542 treatment) of Nodal signaling activity (Supplementary Fig. 9i), we suspected that the Nodal-signaling-mediated promotion of glucose uptake is through unknown glucose metabolism genes, and *Nemep* may be one of them.”

9. Considering the current knowledge that Akt is activated by TGF- β -related proteins and that Akt can promote glucose uptake, the authors should evaluate the effects of Akt inhibition in an assay like Fig 7a.

Response: We thank the reviewer for pointing this out. In order to answer reviewer’s question, we first investigated whether Nodal signaling activates AKT during mesendoderm differentiation. We have now performed experiments and shown in Response Document Fig. II (for reviewer only). We measured the levels of AKT phosphorylation at Thr308 and Ser473 under different condition as indicated. There is no obvious difference for AKT phosphorylation at Thr308 and Ser473 in Activin treated or non-treated day 3 EBs. And also, the levels of Akt phosphorylation at Thr308 and Ser473 in *Cripto/Cryptic* KD cells, *Smad2* KO/*Smad3* KD, and *Trim33* KO cells are not altered compared to WT cells. These results demonstrate that Nodal signaling does not affect AKT phosphorylation during mesendoderm differentiation. In addition, we showed NEMEP KO does not affect AKT phosphorylation at Thr308 and Ser473 (Supplementary Fig. 9g).

Response Document Fig. II

A. Immunoblotting analysis of indicated proteins in non-treated or Activin-treated for 2 hours day 3 EBs.

B. Immunoblotting analysis of indicated proteins in WT or *Smad2* KO/*Smad3* KD day 3 EBs.

C. Immunoblotting analysis of indicated proteins in WT or *Cripto* KD/*Cryptic* KD day 3 EBs.

D. Immunoblotting analysis of indicated proteins in WT or *Trim33* KO day 3 EBs. GAPDH was used as internal control.

10. central conclusion of this manuscript. However, the data clearly argue against this conclusion! In the absence of NEMEP, the fold induction in response to activin remains the same as in control cells. The overall levels are lower in both control and activin-treated EBs. So, I would argue that NEMEP may enhance the basal and induced levels, and thus facilitates glucose import, but does not account for the activin- or nodal-induced increase.

Response: We thank the reviewer for focusing our attention here. Given that NEMEP KO does not affect Nodal signaling activity (Fig. 3c) and the fold induction of glucose uptake in response to Activin remains the same as in control cells (Fig. 7e), we highly suspected that Nodal signaling may induce other unknown genes regulating glucose uptake, which are worthy to be explored in the near future. These unknown genes might also help cells in response to Nodal signaling to regulate glucose uptake. We made corresponding changes in the revised manuscript (lines 390-404).

Taken together, I do agree with the roles of glucose and NEMEP in mesendoderm differentiation, that NEMEP expression is induced in response to Nodal, and that it interacts with GLUT1 and GLUT3. I do, however, not agree that NEMEP accounts for the activin/nodal-induced enhancement of glucose import, since the fold induction is not affected by the absence of NEMEP. Rather, it seems that NEMEP may have a role as facilitator of glucose import, likely by cooperating with other facilitators, some of which may be induced by Smad2/3 activation. Whether this facilitating role occurs through association with GLUT1 and/or GLUT3 is not known but may be assumed.

Response: We thank the reviewer for focusing our attention here. We made corresponding changes in the manuscript title, subtitle (lines 332-333) and result part (lines 390-404) in the revised manuscript.

Minor:

11.- Abstract: The TGF- β superfamily is by its very definition a family, not a superfamily. Many scientists propagate the name superfamily, even though they should not.

Response: Thank you for pointing this out. We made changes accordingly throughout the revised manuscript.

12.- line 150, last word: "the" should be "a".

Response: Thank you for pointing this out. We made changes accordingly in the revised manuscript.

Let us again thank the reviewer for their positive and detailed assessment of our work and for their thoughtful and constructive comments that have helped us improve our manuscript significantly.

REVIEWER COMMENTS

Reviewer #1 (Remarks to the Author):

Thank you for carrying out the knock-down experiment. Although this experiment displays the importance of GLUT1 and GLUT3 in mesendoderm differentiation, it does not support the conclusion that NEMEP facilitates glucose uptake of GLUT1 and GLUT3 through direct interaction. Double knock-down of GLUT1 and GLUT3 must have substantial physiological effect to mESCs, which has active glucose metabolism. I think that my prior concern ( I would like to know how the authors have interpreted the finding that high glucose condition could not compensate the lack of NEMEP for mesendoderm differentiation. If NEMEP acts solely as an activator of GLUT1 and GLUT3, high glucose condition should be sufficient for successful mesendoderm differentiation.) still remains unanswered.

Reviewer #2 (Remarks to the Author):

Fu et al. (Xi) Nature Communications RE-REVISED

The authors describe the identification of a lnc RNA that encodes a small transmembrane peptide that they call NEMEP (Nodal enhanced mesendoderm micropeptide) and propose its expression to be directly induced in response to Nodal signaling through Smad2 and/or Smad3. They carry out a set of experiments that lead them to propose a model whereby, in response to Nodal, NEMEP is expressed and is required for mesendoderm differentiation. They show that NEMEP associates with the glucose transporters Glut1 and Glut3, and propose that NEMEP promotes glucose uptake through its interaction with Glut1 and Glut3, and that this occurs in response to Nodal signaling. They also causally link the requirement of NEMEP in mesendoderm differentiation to Nodal-induced, NEMEP-dependent glucose uptake.

The revised manuscript has further improved when compared to the previous, revised version. In re-revising the manuscript, the authors added some data and, in line with my previous concerns, softened their previous strong conclusions about the mechanism. However, while this weakening of mechanistic conclusions is apparent in the rebuttal and some statements in the manuscript, the authors maintain the previous conclusions in places where they are most visible, e.g. abstract and subtitles. This relates primarily to the “requirement” of NEMEP for nodal-induced glucose uptake and the “required interaction” of NEMEP with Glut1 and/or Glut3 for NEMEP-induced glucose-uptake.

As stated in my previous report, the proposed scenario for a molecular mechanism combines the following elements: (1) Nodal induces, through directed effects of Smad2 and/or Smad3, the expression of a lncRNA encoding NEMEP, (2) NEMEP associates with Glut1 and Glut3, (3) NEMEP controls and is required for increased glucose uptake in response to Nodal, through its association with Glut1 or Glut3. They additionally propose that (4) NEMEP is required for mesendoderm differentiation, and that (5) this requirement for mesendoderm differentiation is explained by NEMEP’s role in glucose uptake through Glut1 and Glut3.

The conclusions that remain not supported are: (1) NEMEP acts “through” its association with Glut1 and/or Glut3, (2) NEMEP is “required” for glucose uptake. Let me be more specific:

1. The Abstract states “we show that NEMEP promotes glucose uptake through its interactions with ... GLUT1 and GLUT3 (lines 25-26). The end of Introduction states that “NEMEP ... promotes glucose uptake through these physical interactions” (lines 83-84). The title on lines 258-259 reads “NEMEP facilitates glucose uptake through its interaction with GLUT1 and GLUT3. Lines 364-365 reads “our finding that the NEMEP-GLUT1/GLUT3 physical interaction synergistically promotes glucose uptake”. Lines 378-379 reads “NEMEP likely functions by interacting with GLUT1 and GLUT3”. In Discussion, lines 411-413 state that “the present study demonstrates that physical interaction of NEMEP with GLUT1/GLUT3 ... augments glucose uptake”.

All these statements are based on the observations that overexpressed NEMEP interacts with Glut1 or Glut3 (non-functional assays, Fig. 5a-c) and that some mutants of NEMEP that do not interact with Glut1/3 do not enhance glucose import (Fig. 5h-j). The experimental basis for this conclusion is insufficient. I am most concerned about the use in Fig. 5i, j of the NEMEP mutant that lacks its TMD as a basis to conclude that the interaction of NEMEP with Glut1/3 is required for NEMEP’s ability to enhance glucose import. I am not sure that the mutant is expressed since the GFP fusion of this mutant seems to have the same size as GFP itself, but, most importantly, the lack of a TMD ensures that it cannot be inserted in the membrane, thus predicting non-functionality. The use of the Delta 51-57 mutant might have been more informative. Hence, there is no convincing evidence that NEMEP acts THROUGH its interaction with GLUT1 and/or GLUT3.

2. Title on line 313 reads “NEMEP is required for glucose uptake...”. Line 370 concludes “NEMEP is required for glucose uptake”.

This conclusion stands in contrast to the data showing that knockout of NEMEP lowers only to some extent the glucose uptake and does not affect the fold induction of glucose import in response to activin (and by extension Nodal). Hence, it is NOT required. It is fine, however, to conclude that NEMEP facilitates glucose import.

Repeating the conclusion of my review of the previous version:

Taken together, I do agree with the roles of glucose and NEMEP in mesendoderm differentiation, that NEMEP expression is induced in response to Nodal, and that it interacts with GLUT1 and GLUT3. NEMEP does not appear to mediate the activin/nodal-induced enhancement of glucose import, since the fold induction is not affected by the absence of NEMEP. Rather, it seems that NEMEP may facilitate glucose import, likely by cooperating with other facilitators, some of which may be induced by Smad2/3 activation. Whether this facilitating role occurs through association with GLUT1 and/or GLUT3 is not known but may be assumed (but cannot be concluded).

Minor:

- line 193: “protein with a molecular weight of about 7 KD” refers to Fig. 2h, where I see a protein marked as 17 KD.
- lines 360-361: Please note that, besides as yet unknown genes, other signaling events could be invoked. For example, phosphorylation of GLUT1 or GLUT3 might enhance glucose import. Direct, TGF- β -induced phosphorylation changes of a large number of proteins have been demonstrated through extensive phosphoproteome analysis (Science Signaling 2014).

REVIEWER COMMENTS:

Reviewer #1 (Remarks to the Author):

Thank you for carrying out the knock-down experiment. Although this experiment displays the importance of GLUT1 and GLUT3 in mesendoderm differentiation, it does not support the conclusion that NEMEP facilitates glucose uptake of GLUT1 and GLUT3 through direct interaction. Double knock-down of GLUT1 and GLUT3 must have substantial physiological effect to mESCs, which has active glucose metabolism. I think that my prior concern (I would like to know how the authors have interpreted the finding that high glucose condition could not compensate the lack of NEMEP for mesendoderm differentiation. If NEMEP acts solely as an activator of GLUT1 and GLUT3, high glucose condition should be sufficient for successful mesendoderm differentiation.) still remains unanswered.

Response: We thank the reviewer for focusing our attention here. We have now updated our revised Fig. 5i,j to support that NEMEP facilitates glucose uptake, likely through interaction with GLUT1 and GLUT3. However, it does not exclude the possibility that NEMEP might facilitate the glucose uptake via other factors (which could also be regulated by Nodal signaling) (Line 292-Line 295; Line 352-362). In addition to the discussion (Line 420-Line 424), we also added the contents (Line 304-Line 307) in the revised manuscript to incorporate our current interpretation of the high glucose condition result noted by the Reviewer. Our current thinking can be explicated in three parts:

i) The process of mesendoderm differentiation is tightly regulated in a spatial and temporal manner (PMID: 23217421, PMID: 29153705, PMID: 27328872, PMID: 27328871): the gene regulatory networks that dictate mesendoderm specification are under the control of distinct combinations of lineage determined transcription factors (LDTFs) that are necessary to elicit cell fate and lineage determination. The gene expression and chromatin accessibility state of the cell changes substantially throughout mesendoderm differentiation. In our *in vitro* differentiation assay, the expression of LDTFs change dynamically, (i.e., not in any consistent pattern for this set of genes) (Fig. 1e,d). In addition, it is well known that a metabolic switch occurs upon mESCs differentiation *in vitro*: specifically, differentiating ESCs downregulate the glycolysis and oxidize most of the glycolysis-derived pyruvate present in mitochondria via oxidative-phosphorylation (OXPHOS) (PMID: 25738450, PMID: 25738455). It is therefore clear that the carbon metabolism occurring during mesendoderm differentiation is complex, involving multiple energy cycles and pathways. The energy requirement for the dynamic mesendoderm differentiation process may be regulated at a spatial and/or temporal level. Accordingly, we cannot assume that providing a constant concentration of glucose would yield informative biological insights about the specific

impacts of glucose on particular mesendoderm differentiation defects caused by depletion of NEMEP.

Indeed, our *in vitro* differentiation assays revealed that both decreasing the glucose concentration and increasing the glucose concentration in the culture medium led to reduced expression of mesendoderm developmental marker genes (including LDTFs like *Mixl1* and *Gsc*) in WT EBs (Fig. 6a,d). It bears emphasis that our CRISPR-dCas9-VP64 (CRISPRa) mediated transcriptional activation of NEMEP results show the decreased expression of LDTFs, and overexpression of GLUT1 or GLUT3 also inhibits the transcription of LDTFs during mesendoderm differentiation (Response Document Figure I. a-c), suggesting that normal mesendoderm differentiation requires a developmentally-appropriate (and potentially regulated) supply of glucose. Given the expression levels of NEMEP and GLUT1 or GLUT3 are gradually induced during mesendoderm differentiation (Fig. 1d, Supplementary Fig. 9h), it is possible that the rate of glucose uptake is gradually increased during mesendoderm differentiation, perhaps in an NEMEP-related manner.

Response Document Figure I

a, WT mESCs or mESCs with CRISPR-dCas9-VP64 (CRISPRa) mediated transcriptional activation of NEMEP expression were induced for EB formation for the indicated durations, and RNA extracts from these cells were analyzed by qPCR using primers for the indicated genes.

b, Validating *Slc2a1* or *Slc2a3* expression in WT and GLUT1 or GLUT3- overexpressing mESCs by qPCR.

c, WT mESCs or GLUT1- or GLUT3-overexpressing mESCs were induced for EB formation for the indicated durations, and RNA extracts from these cells were analyzed by qPCR using primers for the indicated genes.

ii) It bears emphasis that we did not claim NEMEP acts solely as an activator (facilitator) of GLUT1 and GLUT3. It has been reported that other, similarly size peptides such as SPAR have at least two physiologically distinct functions in muscle regeneration (PMID: 28024296) and endothelial fate specification (PMID: 31990292). Thus, it is not implausible to speculate that NEMEP in mESCs could have more than one biomolecular function.

Indeed, our data for NEMEP-GFP pull-down followed by mass-spec analysis in EBs at day 3 not only revealed two glucose transporters (GLUT1 and GLUT3) among the top-ranking candidate NEMEP-interacting proteins, it also identified additional NEMEP binding proteins (e.g., ATP1B1) (Response Document Figure II, only for reviewer), which could in theory affect mesendoderm differentiation.

Response Document Figure II: a, Physical interactions of mouse NEMEP with mGLUT1, mGLUT3 and mATP1B1. Lysates from HEK293T cells co-transfected with plasmids encoding mGLUT1-FLAG or mGLUT3-FLAG or mATP1B1-FLAG or control vector and mNEMEP-HA (as indicated) were immunoprecipitated with anti-FLAG affinity beads, and immune complexes were analyzed by immunoblotting using an antibody against HA. The protein inputs were detected with western blotting using antibodies against FLAG and HA from same amount of cell lysates; **b**, Physical interactions of human NEMEP with hNEMEP. Lysates from HEK293T cells co-transfected with plasmids encoding hNEMEP-FLAG or control vector and hNEMEP-HA (as indicated) were immunoprecipitated with anti-FLAG affinity beads, and immune complexes were analyzed by immunoblotting using an antibody against HA. The protein inputs were detected by western blotting using antibodies against FLAG and HA from same amount of cell lysates.

iii) Finally, we would like to make a distinction between the proposed biochemical function (*i.e.*, NEMEP affecting glucose uptake by transporter proteins) versus the overall impact of NEMEP on the mesendoderm differentiation process. There could be emergent physiological events with regulatory impacts (*e.g.*, feedback or feedforward inhibitory loops, etc.) that remain presently obscure. It is premature to exclude such possibilities, and simply providing a constant concentration of glucose to NEMEP knock-out mESCs throughout mesendoderm differentiation would not necessarily represent or appropriately compensate (recapitulate) the glucose requirement of these highly dynamic and tightly regulated developing cells.

Let us again thank the reviewer for the helpful guidance to improve our study.

Reviewer #2 (Remarks to the Author):

Fu et al. (Xi) Nature Communications RE-REVISED

The authors describe the identification of a lnc RNA that encodes a small transmembrane peptide that they call NEMEP (Nodal enhanced mesendoderm micropeptide) and propose its expression to be directly induced in response to Nodal signaling through Smad2 and/or Smad3. They carry out a set of experiments that lead them to propose a model whereby, in response to Nodal, NEMEP is expressed and is required for mesendoderm differentiation. They show that NEMEP associates with the glucose transporters Glut1 and Glut3, and propose that NEMEP promotes glucose uptake through its interaction with Glut1 and Glut3, and that this occurs in response to Nodal signaling. They also causally link the requirement of NEMEP in mesendoderm differentiation to Nodal-induced, NEMEP-dependent glucose uptake.

The revised manuscript has further improved when compared to the previous, revised version. In re-revising the manuscript, the authors added some data and, in line with my previous concerns, softened their previous strong conclusions about the mechanism. However, while this weakening of mechanistic conclusions is apparent in the rebuttal and some statements in the manuscript, the authors maintain the previous conclusions in places where they are most visible, e.g. abstract and subtitles. This relates primarily to the “requirement” of NEMEP for nodal-induced glucose uptake and the “required interaction” of NEMEP with Glut1 and/or Glut3 for NEMEP-induced glucose-uptake. As stated in my previous report, the proposed scenario for a molecular mechanism combines the following elements: (1) Nodal induces, through directed effects of Smad2 and/or Smad3, the expression of a lncRNA encoding NEMEP, (2) NEMEP associates with Glut1 and Glut3, (3) NEMEP controls and is required for increased glucose uptake in response to Nodal, through its association with Glut1 or Glut3. They additionally propose that (4) NEMEP is required for mesendoderm differentiation, and that (5) this requirement for mesendoderm differentiation is explained by NEMEP’s role in glucose uptake through Glut1 and Glut3.

The conclusions that remain not supported are: (1) NEMEP acts “through” its association with Glut1 and/or Glut3, (2) NEMEP is “required” for glucose uptake. Let me be more specific:

1. The Abstract states “we show that NEMEP promotes glucose uptake through its interactions with ... GLUT1 and GLUT3 (lines 25-26). The end of Introduction states that “NEMEP ... promotes glucose uptake through these physical interactions” (lines 83-84). The title on lines 258-259 reads “NEMEP facilitates glucose uptake through its interaction with GLUT1 and GLUT3. Lines 364-365 reads “our finding that the NEMEP-GLUT1/GLUT3 physical interaction synergistically promotes glucose uptake”. Lines 378-379 reads “NEMEP likely functions by interacting with GLUT1 and GLUT3”. In Discussion, lines 411-413 state that “the present study demonstrates that physical interaction of NEMEP with GLUT1/GLUT3 ... augments glucose uptake”.

All these statements are based on the observations that overexpressed NEMEP interacts with Glut1 or Glut3 (non-functional assays, Fig. 5a-c) and that some mutants of NEMEP that do not interact with Glut1/3 do not enhance glucose import (Fig. 5h-j). The experimental basis for this conclusion is insufficient. I am most concerned about the use in Fig. 5i, j of the NEMEP mutant that lacks its TMD as a basis to conclude that the interaction of NEMEP with Glut1/3 is required for NEMEP’s ability to enhance glucose import. I am not sure that the mutant is expressed since the GFP fusion of this mutant seems to have the same size as GFP itself, but, most importantly, the lack of a TMD ensures that it cannot be inserted in the membrane, thus predicting non-functionality. The use of the Delta 51-57 mutant might have been more informative. Hence, there is no convincing evidence that NEMEP acts THROUGH its interaction with GLUT1 and/or GLUT3.

Response: We would like to thank the reviewer for the careful review and for the excellent guidance about how to improve our study.

__We thank the reviewer’s excellent suggestion to conduct experiments using the NEMEP-Δ51-57 mutant. During the second revision round, we actually completed an experiment using the NEMEP-Δ51-57 mutant as a control, but did not include this data in Fig. 5i, j. We have now added this data in the newly revised Fig. 5i, j as described in the revised manuscript:

“Intriguingly, we detected NEMEP-mediated enhancements on both glucose uptake and glycolysis activities in mESCs in experiments examining overexpression of GLUT1 or GLUT3. Specifically, the expression of WT NEMEP but not TMD-deletion NEMEP or H51-F57-deletion NEMEP in mESCs overexpressing GLUT1 or GLUT3 resulted in significantly higher glucose

uptake compared to the GLUT1 or GLUT3 overexpressing mESCs (Fig. 5i, j). Our finding that the H51-F57-deletion variant of NEMEP failed to form a complex with GLUT1 or GLUT3 (Fig 5d) suggests that NEMEP may facilitate glucose uptake through interaction with GLUT1 or GLUT3. In addition, the expression of WT NEMEP but not TMD-deletion NEMEP in mESCs overexpressing GLUT1 or GLUT3 resulted in significantly higher glycolysis activities compared to the GLUT1 or GLUT3 overexpressing mESCs (Supplementary Fig. 8a-h). Therefore, NEMEP's facilitation of glucose uptake seems likely to occur through its interactions with GLUT1 and GLUT3. However, it does not exclude the possibility that NEMEP might facilitate the glucose uptake via other factors (which could also be regulated by Nodal signaling).”

Fig. 5i and 5j: Glucose uptake analysis in HEK293T cells expressing plasmids for overexpression of the indicated proteins or protein pairs. The values are normalized to the total protein concentration of each sample (means ± S.E.M., n = 3).

Our Sanger sequencing analysis validated that the examined fusion proteins (NEMEP-ΔTMD-GFP and NEMEP-del-51-57-GFP) are in the correct reading frame (Response Document Figure III).

a

b

c

d

Response Document Figure III

- a, Validation of NEMEP-ΔTMD-GFP plasmid by Sanger sequencing.
- b, Amino acid sequence of the NEMEP-ΔTMD-GFP fusion protein.
- c, Validation of the NEMEP-Δ51-57-GFP plasmid by Sanger sequencing.
- d, Amino acid sequence of the NEMEP-Δ51-57-GFP fusion protein.

__ We have examined the subcellular localizations of the WT-NEMEP-GFP, NEMEP-ΔTMD-GFP, and NEMEP-Δ51-57-GFP fusion proteins using GFP-imaging to establish that NEMEP-Δ51-57 is localized in the membrane like WT NEMEP; NEMEP-ΔTMD is not membrane localized (Response Document Figure IV).

Response Document Figure IV: Images of WT-NEMEP-GFP, NEMEP- Δ TMD-GFP and NEMEP- Δ 51-57-GFP in HEK293T cells. Nuclei were stained with Hoechst33342. Scale bar, 10 μ m.

__In addition, our data in Supplementary Fig. 8d,h clearly show that NEMEP- Δ TMD-GFP migrates between NEMEP-GFP and GFP alone.

8d

8h

Supplementary Fig. 8d and 8h: Lysates from HEK293T cells expressing plasmids for expression of the indicated proteins or protein pairs were analyzed by immunoblotting using antibodies against FLAG, GFP, and GAPDH; * indicates NEMEP- Δ TMD-GFP.

__The expression of NEMEP- Δ TMD-GFP is shown in Fig 5d in both the left and right subpanels. Note that the GFP fusion protein (NEMEP- Δ TMD-GFP) migrates slower than GFP alone in the right subpanel, but not in the left subpanel, consistent with the size of the mutant variant being

around 4.6 KD larger than the GFP protein (MW=26.8 KD). The run time for the gel presented in the left subpanel was apparently insufficient to resolve these two similarly sized of proteins (GFP and NEMEP- Δ TMD-GFP).

2. Title on line 313 reads “NEMEP is required for glucose uptake...”. Line 370 concludes “NEMEP is required for glucose uptake”.

This conclusion stands in contrast to the data showing that knockout of NEMEP lowers only to some extent the glucose uptake and does not affect the fold induction of glucose import in response to activin (and by extension Nodal). Hence, it is NOT required. It is fine, however, to conclude that NEMEP facilitates glucose import.

Response: We thank the reviewer for correcting this error in our reasoning. We have made appropriate changes throughout the revised manuscript by using “facilitates glucose uptake” in the abstract, and subheadings (Please see below):

ABSTRACT (Line 24-Line 26): “We show that NEMEP interacts with the glucose transporters GLUT1/GLUT3 and facilitates glucose uptake, likely through these interactions”

SUBHEADINGS

Line 239: “NEMEP interacts with GLUT1/GLUT3 and facilitates glucose uptake.”

Line 296: “NEMEP facilitates glucose uptake during mesendoderm differentiation.”

Repeating the conclusion of my review of the previous version:

Taken together, I do agree with the roles of glucose and NEMEP in mesendoderm differentiation, that NEMEP expression is induced in response to Nodal, and that it interacts with GLUT1 and GLUT3. NEMEP does not appear to mediate the activin/nodal-induced enhancement of glucose import, since the fold induction is not affected by the absence of NEMEP. Rather, it seems that NEMEP may facilitate glucose import, likely by cooperating with other facilitators, some of which may be induced by Smad2/3 activation. Whether this facilitating role occurs through association with GLUT1 and/or GLUT3 is not known but may be assumed (but cannot be concluded).

Response: We now appreciate the previous flaws in our reasoning, and have—under the excellent and patient guidance of the reviewer—now corrected the flaws in our argument in the

revised manuscript (with attendant changes in the abstract, introduction, subheadings, results, and discussion sections. Additionally, happily, the suggestion to use the NEMEP- Δ 51-57 mutant also enabled us to clarify some previously ambiguous inferences.

As examples of the related revisions:

INTRODUCTION:

Line 76-Line 77: “we discovered that NEMEP interacts with two glucose transporters (GLUT1 and GLUT3) and facilitates glucose uptake.”

RESULTS:

Line 282-Line 295: “Intriguingly, we detected NEMEP-mediated enhancements on both glucose uptake and glycolysis activities in mESCs in experiments examining overexpression of GLUT1 or GLUT3. Specifically, the expression of WT NEMEP but not TMD-deletion NEMEP or H51-F57-deletion NEMEP in mESCs overexpressing GLUT1 or GLUT3 resulted in significantly higher glucose uptake compared to the GLUT1 or GLUT3 overexpressing mESCs (Fig. 5i, j). Our finding that the H51-F57-deletion variant of NEMEP failed to form a complex with GLUT1 or GLUT3 (Fig 5d) suggests that NEMEP may facilitate glucose uptake through interaction with GLUT1 or GLUT3. In addition, the expression of WT NEMEP but not TMD-deletion NEMEP in mESCs overexpressing GLUT1 or GLUT3 resulted in significantly higher glycolysis activities compared to the GLUT1 or GLUT3 overexpressing mESCs (Supplementary Fig. 8a-h). Therefore, NEMEP’s facilitation of glucose uptake seems likely to occur through its interactions with GLUT1 and GLUT3. However, it does not exclude the possibility that NEMEP might facilitate the glucose uptake via other factors (which could also be regulated by Nodal signaling).”

Line 357-Line 362: “That is, our results support a model of mesendoderm differentiation from pluripotent mESCs wherein activated Nodal signaling induces the expression of a micropeptide, NEMEP that may function by interacting with GLUT1 and GLUT3 or (other proteins involved in glucose uptake), which could selectively modulate glucose uptake to meet the specific energy needs throughout the tightly regulated mesendoderm differentiation process (Fig. 7g).”

DISCUSSION:

Line 389-Line 391: “Our work in the present study demonstrates that NEMEP interacts with the GLUT1/GLUT3 glucose transporter proteins and augments glucose uptake during mesendoderm differentiation.”

Minor:

3.- line 193: “protein with a molecular weight of about 7 KD” refers to Fig. 2h, where I see a protein marked as 17 KD.

Response: We thank the reviewer for pointing out this error. We have corrected this in revised Fig. 2h.

4.- lines 360-361: Please note that, besides as yet unknown genes, other signaling events could be invoked. For example, phosphorylation of GLUT1 or GLUT3 might enhance glucose import. Direct, TGF- β -induced phosphorylation changes of a large number of proteins have been demonstrated through extensive phosphoproteome analysis (Science Signaling 2014).

Response: We thank the reviewer for focusing our attention here. We have added content in the results section of the revised manuscript (Line 351-Line 355) and cited the paper (Science Signaling 2014):

“we suspect that: i) Nodal signaling induces the transcriptional activation of other, as-yet-unknown genes that somehow regulate glucose uptake; ii) crosstalk between Nodal signaling and other signaling pathways may occur to facilitate glucose uptake; and iii) Nodal signaling may modulate post-translational modifications (e.g., phosphorylation) of proteins involved in facilitating glucose uptake (Science Signaling 2014).”

Let us again thank the reviewer for their positive and detailed assessment of our work and for their thoughtful and constructive comments that have helped us improve our manuscript significantly.

REVIEWERS' COMMENTS

Reviewer #1 (Remarks to the Author):

Although the question whether interaction of NEMEP with GLUT1 or GLUT3 in fact activates their glucose transport activity directly still remains uncertain and I am curious to know, the wording

We show that NEMEP interacts with the glucose transporters GLUT1/GLUT3 and facilitates glucose uptake, likely through these interactions.

is consistent with the observations.

Reviewer #2 (Remarks to the Author):

This is the third revision of the manuscript. The authors have (finally) addressed the comments and modified their conclusions to appropriately reflect their results. I have no further comments.

REVIEWER COMMENTS:

Reviewer #1 (Remarks to the Author):

Although the question whether interaction of NEMEP with GLUT1 or GLUT3 in fact activates their glucose transport activity directly still remains uncertain and I am curious to know, the wording

We show that NEMEP interacts with the glucose transporters GLUT1/GLUT3 and facilitates glucose uptake, likely through these interactions.

is consistent with the observations.

Response: We thank the reviewer to bring our attention to this point. This sentence is the summary for results of Fig. 5 and extended data Fig. 6-8 (described below), which is consistent with our observation. We actually took the excellent suggestion from reviewer #2 to include the NEMEP-Δ51-57 mutant in Fig. 5i,j (2nd revision) during last round of revision. We added this data in the revised Fig. 5i, j as described in the revised manuscript (3rd revision) (highlighted in yellow):

“NEMEP interacts with GLUT1/GLUT3 and facilitates glucose uptake

To explore the molecular mechanism of NEMEP in mesendoderm differentiation, we conducted pull-down experiments with EBs at day 3 using NEMEP-GFP as bait. Mass spectrometry analysis of co-purified proteins revealed two glucose transporters (GLUT1 and GLUT3) among the top-ranking candidate NEMEP-interacting proteins. Co-immunoprecipitation (co-IP) studies validated that both GLUT1 and GLUT3 do physically interact with NEMEP (Fig. 5a). Then, we confirmed that the homologous human NEMEP protein also interacts with the human GLUT1 and GLUT3 proteins (Fig. 5b). GLUT1 to GLUT4 are class I facilitative glucose transporters⁴⁹. *Glut1* and *Glut3* have much higher expression in mESCs compared to *Glut2* or *Glut4* (Supplementary table 1). We conducted bimolecular fluorescence complementation (BiFC) analysis and successfully validated the interaction between NEMEP and these four of class I facilitative glucose transporters (Supplementary Fig. 6a).

Moreover, both confocal microscope images from BiFC assay and transient transfection of both GLUT1-FLAG or GLUT3-FLAG and NEMEP-HA vectors into mESCs show the colocalization of GLUT1 and GLUT3 with NEMEP on the cell membrane (Supplementary Fig. 6b, c). We also validated that NEMEP binds to endogenous GLUT1 and GLUT3 using NEMEP-3xFLAG knock-in mESCs (Fig 5c). Further, deletion of the transmembrane domain (TMD) and of a 7-residue region (lacking H51-F57) of the NEMEP C terminus disrupted the interactions between NEMEP and GLUT1/GLUT3 proteins (Fig. 5d and Supplementary Fig. 6d). Domain swapping of the NEMEP-TMD domain for the ACVR1-TMD or ITGB1-TMD also impaired the interactions (Fig. 5d). These results demonstrate that the TMD domain and NEMEP residues H51-F57 are essential for NEMEP-GLUT1 and -GLUT3 interactions.

We next examined whether NEMEP impacts glucose transport by using the CRISPR-dCas9-VP64 activator system (CRISPRa) to generate mESCs that overexpress the endogenous mRNA encoding NEMEP (*Gm11549*) (Supplementary Fig. 7a). Compared to WT mESCs, we found that overexpression of NEMEP resulted in a ~30% increase in glucose consumption as well as a ~60% increases in lactate excretion (Fig. 5e, f). Consistently, a Seahorse-based analysis measuring the extracellular acidification rate (ECAR) showed that the glycolysis activity of the NEMEP-overexpressing mESCs was significantly higher than in WT mESCs (Fig. 5g and Supplementary Fig.

7b, c). Moreover, the overexpression of human NEMEP in HepG2 cells (Supplementary Fig. 7d) also significantly increased glucose consumption, lactate excretion, glycolysis activity (Supplementary Fig. 7e-g).

We then used Promega Glucose Uptake-Glo™ Assays to measure mESC glucose uptake to examine whether NEMEP impacts glucose transporter activity. Note that the measurement readout of the assay specifically reflects the glucose transporter activity. No changes in transporter activity were detected in mESCs expressing diverse NEMEP mutant variants (including TMD deletion, H51-F57 deletion, and the domain-swapped ACVR1-TMD and ITGB1-TMD variants); note that we did detect the expected increase in glucose transporter activity in mESCs expressing the full-length, wild type NEMEP (Fig. 5h).

Intriguingly, we detected NEMEP-mediated enhancements on both glucose uptake and glycolysis activities in mESCs in experiments examining overexpression of GLUT1 or GLUT3. Specifically, the expression of WT NEMEP but not TMD-deletion NEMEP or H51-F57-deletion NEMEP in mESCs overexpressing GLUT1 or GLUT3 resulted in significantly higher glucose uptake compared to the GLUT1 or GLUT3 overexpressing mESCs (Fig. 5i, j). Our finding that the H51-F57-deletion variant of NEMEP failed to form a complex with GLUT1 or GLUT3 (Fig. 5d) suggests that NEMEP may facilitate glucose uptake through interaction with GLUT1 or GLUT3. In addition, the expression of WT NEMEP but not TMD-deletion NEMEP in mESCs overexpressing GLUT1 or GLUT3 resulted in significantly higher glycolysis activities compared to the GLUT1 or GLUT3 overexpressing mESCs (Supplementary Fig. 8a-h). Therefore, NEMEP's facilitation of glucose uptake seems likely to occur through its interactions with GLUT1 and GLUT3. However, it does not exclude the possibility that NEMEP might facilitate the glucose uptake via other factors (which could also be regulated by Nodal signaling)."

Fig. 5i and 5j: Glucose uptake analysis in HEK293T cells expressing plasmids for overexpression of the indicated proteins or protein pairs. The values are normalized to the protein concentration (means ± S.E.M., n = 3 biological independent samples).

Let us again thank the reviewer for their positive and detailed assessment of our work and for their thoughtful and constructive comments that have helped us improve our manuscript significantly.